# Persistence and turnover of soil organic carbon in global drylands

Hui Wang [1,2], Fernando T. Maestre [3], Nan Lu [4,5,6] ✉, Guang Zhao [7], Yangjian Zhang [7,8], Sergio Asensio [9], De Shorn E. Bramble [1], Weiliang Chen [4,10], Michaela A. Dippold [2], David J. Eldridge [11], Juan J. Gaitán [12], Miguel García-Gómez [13], Beatriz Gozalo [9], Nicolas Gross [14], Emilio Guirado [3], Yoann Le Bagousse-Pinguet [15], Jaime Martínez-Valderrama [9,16], Betty J. Mendoza [17], Victoria Ochoa [18], César Plaza [12,19], Hugo Saiz [20], Marion Schrumpf [1], Carlos A. Sierra [1], Andrés Tangarife-Escobar[1], Enrique Valencia [21], Sophie F. von Fromm [22], Cong Wang [4,5], Kai Wang [1,4], Yunqiang Wang [5,23], Sönke Zaehle [24], Bojie Fu [4,5,6], Susan Trumbore [1] & Jianbei Huang [1] ✉

Reliable predictions of dryland carbon fluxes require understanding the persistence and turnover of soil organic carbon (SOC). We measure radiocarbon to quantify the age of SOC and $CO_2$ released from soil respiration at 97 dryland sites across six continents. Here we show that bulk SOC contains little C fixed in the past 60 years, while respired $CO_2$ originates from both bomb-derived recent C and millennia-old C, challenging the idea that old C is chemically or physically protected. Radiocarbon suggests mean ages of ~2100 years for bulk SOC and ~520 years for respired $CO_2$, the latter far older than machine-learning (<50 years) or Earth system models predict. Aridity, net primary productivity, and SOC content are dominant predictors for radiocarbon signatures, with abrupt shifts to older C beyond an aridity threshold of ~0.87. Our findings underscore the need to incorporate the vulnerability of older carbon into models and land management strategies.

Drylands, regions where the aridity index (the ratio of precipitation to potential evapotranspiration) is below 0.65[1], constitute the largest set of terrestrial biomes on Earth[2] and cover about 41% of the global land area[3]. They are key regulators of the global carbon (C) cycle[4,5], storing approximately 240 P`g C to a depth of 1 m globally[6] and dominating the interannual variability and long-term trend of the terrestrial C cycle[7,8]. Variations in precipitation and temperature influence the dynamics of soil organic carbon (SOC), and thus soil $CO_2$ fluxes, by affecting the amount and quality of fresh C inputs, soil properties, and microbial decomposition[9–12]. However, quantifying how these factors predict the timescales of C persistence, turnover, and vulnerability of SOC remains limited, especially in dryland systems. This knowledge gap leads to significant uncertainties in projections of dryland soil C storage under climate change and land management practices[13,14].

The persistence and turnover of SOC can be assessed using radiocarbon ($^{14}C$)[15]. The natural decay of $^{14}C$ allows estimating the age of C fixed centuries to millennia ago[15], while the penetration of bomb-derived $^{14}C$ produced by the atmospheric testing of thermonuclear weapons in the 1960s can be used for estimating soil C turnover on annual to decadal timescales[16,17]. Recent global syntheses have leveraged bulk $\Delta^{14}C$, i.e., the deviation of a sample's $^{14}C$ content from the atmospheric $^{14}C$ level in 1950, to estimate the mean age of soil C and to examine its control by climatic and mineralogical factors[14,18–20]. The $\Delta^{14}C$ of bulk SOC reflects the age of C dominated by slow pools with C ages of centuries to millennia, commonly thought to be physically protected within aggregates or chemically stabilized on mineral surfaces[21,22]. By contrast, the $\Delta^{14}C$ of respired $CO_2$ reflects the age of C pools that serve as substrates for microbial decomposition during

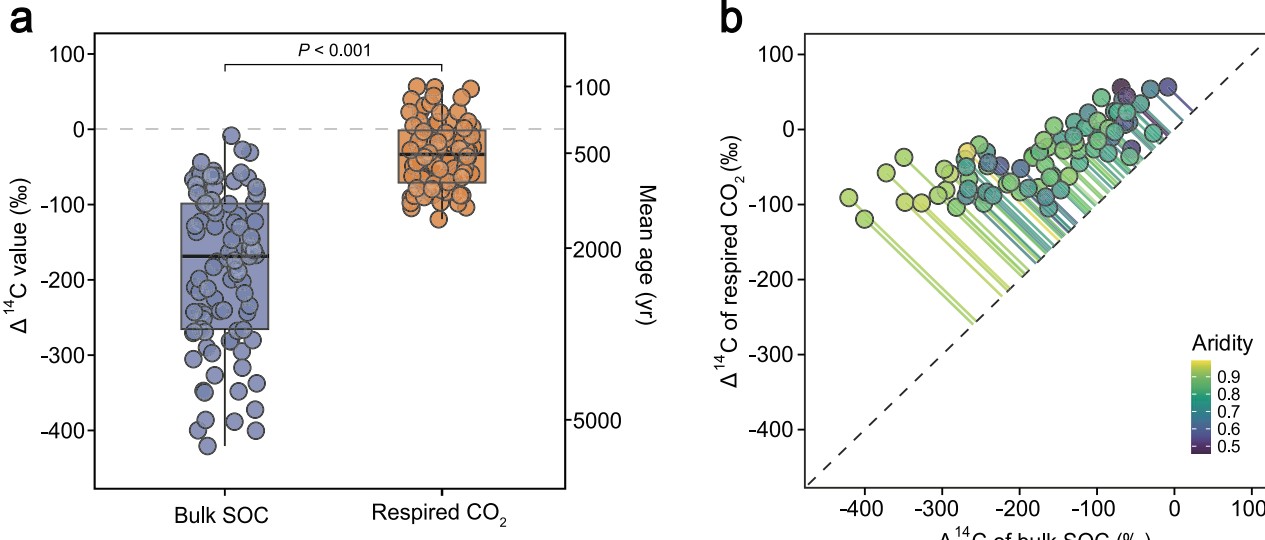

**Fig. 1 | Radiocarbon signatures and mean ages of global dryland soils.** $\Delta^{14}C$ (‰) represents the deviation of a sample's radiocarbon content from the preindustrial atmosphere (0‰). Positive values ($\Delta^{14}C > 0$‰) indicate the presence of bomb-derived $^{14}C$ fixed in the past c. 60 years from atmospheric weapons testing, while negative values ($\Delta^{14}C < 0$‰) indicate C old enough for substantial radioactive decay ($^{14}C$ half-life = 5730 years). **a** $\Delta^{14}C$ of respired $CO_2$ significantly exceeds that of bulk soil organic carbon (SOC), with a corresponding mean $^{14}C$ age of 520 years compared to 2100 years for bulk SOC. Mean ages were calculated from the $\Delta^{14}C$ values using one-pool models. Purple indicates bulk SOC, and orange indicates respired $CO_2$, for both $\Delta^{14}C$ and mean age. Box plots show the median (center line) and the interquartile range (box, from the lower to the upper quartile), with whiskers extending to 1.5 times the interquartile range. Points represent individual observations, and the dashed line indicates 0‰. **b** The $\Delta^{14}C$ difference **b**etween bulk SOC and respired $CO_2$ increases with aridity (1 − aridity index; aridity index = precipitation/potential evapotranspiration)[i]. The dotted black line shows the 1:1 relationship, and colored solid lines indicate the relative magnitude of $\Delta^{14}C$ differences between paired measurements.

incubations[15], and provides a more appropriate benchmark for evaluating model-derived transit times (i.e., the mean time elapsed since the $CO_2$ produced in incubations was fixed from the atmosphere). However, current data syntheses using radiocarbon are biased toward forested regions[19,23], and have less information on $\Delta^{14}C$ of respired $CO_2$, especially from studies designed to encompass broad environmental and geographical gradients. Combining $\Delta^{14}C$ of bulk and respired $CO_2$ can distinguish between more stable, slow-cycling, and active, fast-cycling pools[16,24], thereby providing insights into the mechanisms driving long-term persistence and short-term turnover of SOC in response to changing climatic conditions.

To date, only 13 sites with paired bulk $\Delta^{14}C$ and respired $\Delta^{14}C$ originate from dryland regions (Extended Data Table 1) in the International Soil Radiocarbon Database (ISRaD; http://soilradiocarbon.org)[23]. This hampers our ability to use radiocarbon to constrain carbon–climate feedbacks and reduce uncertainties in model estimates of SOC age[14,19] and turnover time[25,26] in dryland regions. Here, we collected soils from 97 dryland sites spanning six continents and large environmental gradients (Extended Data Fig. 1). We measured $\Delta^{14}C$ of bulk SOC and respired $CO_2$ and analyzed how these values are influenced by climate factors (aridity and mean annual temperature [MAT]), vegetation factors (net primary productivity [NPP], plant cover, and species richness), and soil properties (SOC, pH, fine texture, oxalate-extractable Fe and Al, and microbial respiration). We hypothesize that, with increasing aridity, reduced vegetation inputs and enhanced mineral protection lead to older ages of bulk SOC and respired $CO_2$. We further expect that, although the $\Delta^{14}C$ of both bulk SOC and respired $CO_2$ declines with increasing aridity, the $\Delta^{14}C$ difference between them will widen. This divergence reflects a growing decoupling between respiration, which is increasingly supported by a small pool of relatively fresh plant-derived inputs, and bulk SOC, which is dominated by a larger, older C pool that contributes proportionally less to respiratory fluxes.

## Results and discussion

We found that the $\Delta^{14}C$ values of bulk SOC were consistently negative in global drylands (mean $\Delta^{14}C = -190.0$‰; Fig. 1a), indicating that the SOC resided in soils long enough for significant radioactive decay and had minimal contributions from young, bomb-derived C ($\Delta^{14}C > 0$‰) fixed since the 1960s. While we cannot exclude the possibility of small amounts of $^{14}C$-dead petrogenic OC (−1000‰), the higher $\Delta^{14}C$ values of bulk SOC (−418.0‰ to −5.9‰) prove that petrogenic OC can influence but does not dominate the old SOC pools, comprising at most 16% of the bulk SOC pool (Extended Data Table 2). Across sites, the $\Delta^{14}C$ values of respired $CO_2$ (mean $\Delta^{14}C = -39.2$‰) were consistently higher than those of bulk SOC (Fig. 1a). This suggests that respired $CO_2$ is derived from a mixture weighted toward faster-decomposing substrates, including the labile fraction of bulk SOC (mostly pre-bomb) and bomb-derived recent C, consistent with results from $^{13}C$-labeling studies[27]. Nevertheless, in contrast to wetter regions where respired $CO_2$ is mostly derived from bomb-derived plant C (Extended Data Table 3 and Extended Data Fig. 2), our analysis across dryland sites indicates that, on average, $23 \pm 3\%$ (mean ± SE) of the C used for respiration originates from older C with the $^{14}C$ signature of bulk SOC ($f_{old}$, see "Methods"; Extended Data Table 4). We believe that petrogenic OC is unlikely to be a dominant contributor to respired $CO_2$, because it is highly refractory[28]. Ramped combustion studies typically showed that mineral-associated organic C (MAOC) contains only a small $^{14}C$-dead fraction[29]. The relatively old respired $CO_2$ in drylands could more plausibly be explained by the continuum of ages spanned by biospheric SOC, reflecting physical protection (particulate organic C [POC] vs. MAOC) and variable mineral binding strengths (strong vs. weak)[30,31] (Extended Data Fig. 3). This C has persisted for millennia yet can still be destabilized after rewetting, challenging the long-standing assumption that old SOC in dryland environments is not available for decomposition[32,33].

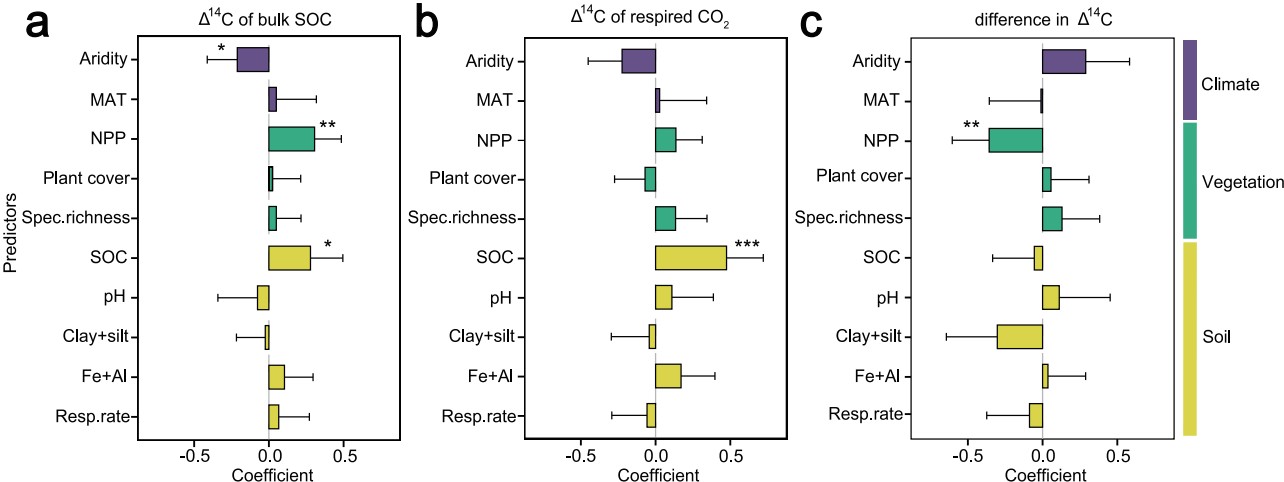

**Fig. 2 | Predictors of radiocarbon signatures in global dryland soils. a** $\Delta^{14}C$ of bulk SOC ($n = 97$, $R^2 = 0.60$). **b** $\Delta^{14}C$ of respired $CO_2$ ($n = 80$, $R^2 = 0.64$). **c** $\Delta^{14}C$ difference between bulk SOC and respired $CO_2$ ($n = 80$, $R^2 = 0.40$). Linear mixed-effects models included climate variables (aridity [1 – aridity index], mean annual temperature [MAT]), vegetation variables (net primary productivity [NPP, ln-transformed], plant cover, and species richness [spec. richness]), and soil properties (SOC, pH, clay + silt content [Clay + silt], oxalate-extractable Fe and Al oxides [Fe + Al], and microbial respiration rate [Resp. rate]). See Extended Data Tables 9–11 for full model results. Error bars show 95% confidence intervals (CIs) of fixed-effect coefficients. Significance levels: *$P < 0.05$, **$P < 0.01$, ***$P < 0.001$. $R^2$ values are conditional, representing variance explained by both fixed and random effects.

When interpreting the $\Delta^{14}C$ of SOC-derived respired $CO_2$, we accounted for potential contributions from $^{14}C$-depleted SIC to the respired $CO_2$ that could bias apparent respiration ages[34,35]. Rewetting soils can promote carbonate dissolution and dissolved inorganic carbon (DIC) formation, resulting in $CO_2$ contributions from both parent-material and pedogenic carbonates[36,37]. For 45 of our sites, SIC content was below 0.1%, and we assumed all respired $CO_2$ was derived from SOC. For soils with SIC content above 0.1%, the potential SIC contribution to respired $CO_2$ ($f_{SIC}$) was estimated using a $\delta^{13}C$-based two-end-member mass balance due to the distinct $\delta^{13}C$ signatures of SIC and SOC (see "Methods"; Extended Data Fig. 4). This estimate was then incorporated into a $\Delta^{14}C$–based mass balance to estimate the influence of SIC on the $\Delta^{14}C$ of SOC-derived, respired $CO_2$. This resulted in the exclusion of 17 sites with high $f_{SIC}$ (mostly >15%) to avoid substantial underestimation of the $\Delta^{14}C$ of SOC-derived respired $CO_2$. Consequently, our analysis and interpretation of $\Delta^{14}C$ of respired $CO_2$ are based on 80 of the 97 sites for which SIC contributions were limited (Extended Data Figs. 5 and 6).

Using a one-pool model, the mean age of bulk SOC in topsoils (<10 cm) across the surveyed dryland sites (which are dominated by grasslands, shrublands, and savannas) was estimated to be 2100 ± 140 years (Fig. 1a; Extended Data Fig. 7; Extended Data Table 5), much older than previous surface soil (<30 cm) age estimates across biomes, including tropical (440 years) and temperate (390 years) forests, grasslands (1200 years), shrublands (680 years), and savannas (510 years)[19]. In addition, previous studies showed that bulk SOC ages generally increased with depth, due to reduced plant C inputs, lower microbial activity, and stronger mineral-associated stabilization in deeper horizons[19,38]. These results suggest that the majority of topsoil SOC in drylands forms over significantly longer timescales than previously thought, indicating greater persistence of bulk SOC but lower incorporation of recent C.

Based on the $\Delta^{14}C$ values of respired $CO_2$ and a one-pool model, the mean transit time was estimated to be 520 ± 30 years (Fig. 1a; Extended Data Fig. 7; Extended Data Table 6). This estimate is similar to a conservative value of 480 ± 50 years based on the contribution of older C pools to respired $CO_2$, but higher than the c. 253 years obtained from fitting a two-pool model to bulk and respired $\Delta^{14}C$ data (see Supplementary Text; Extended Data Tables 6 and 7). In either case, our radiocarbon estimates are far longer than the 37 years derived from current machine-learning approaches (Extended Data Table 6)[26]. Our estimates are also orders of magnitude higher than the year-to-decade turnover times of the fast pool and even the bulk SOC pool in dryland biomes as estimated by ESMs and incubation experiments[25,39]. Global soil turnover time estimates from model- and machine-learning-based approaches remain highly uncertain because they rely on stock-to-flux calculations with poorly constrained inputs (NPP, belowground allocation) or outputs (respiration), as well as assumptions of steady state and SOC homogeneity[26,40]. Our radiocarbon-based estimates provide an empirical constraint and mechanistic understanding of SOC turnover, helping to reduce these uncertainties in global drylands.

We used linear mixed-effects regression models to examine the influence of climate, vegetation, and soil factors on $\Delta^{14}C$ of bulk SOC, $\Delta^{14}C$ of respired $CO_2$, and the differences in $\Delta^{14}C$ of bulk SOC and respired $CO_2$ (Extended Data Tables 8–11). Aridity (1 – [precipitation/potential evapotranspiration]) was more important than MAT (Extended Data Table 9) in predicting variations in bulk $\Delta^{14}C$ in global drylands (Fig. 2a). This result differs from previous global syntheses that identified MAT as the dominant climatic factor, likely because those studies included a wider range of ecosystems but limited dryland data[19]. The effects of temperature on soil $\Delta^{14}C$ through processes like C inputs and SOC decomposition, which are commonly observed in mesic ecosystems[38,41], are diminished in dryland ecosystems due to moisture limitations on microbial activity. Although global syntheses typically use mean annual precipitation (MAP) as the main water availability metric, in our dryland database, aridity not only integrates both precipitation and evapotranspiration but also predicts bulk $\Delta^{14}C$ better than MAP does, so we retained aridity in all linear mixed-effect models. Changes in NPP and SOC content also contributed to observed bulk $\Delta^{14}C$ variations. By contrast, soil properties such as SOC content were found to be more important predictors than climatic and vegetation variables in explaining the variations in $\Delta^{14}C$ of respired $CO_2$ (Fig. 2b and Extended Data Table 10).

Consistent with our hypothesis that reduced vegetation inputs would drive older SOC ages with increasing aridity, we found that aridity led to a decline in bulk $\Delta^{14}C$ (Fig. 3a), reflecting an increase in the mean age of SOC across most regions (Extended Data Fig. 8). Greater SOC persistence in drier regions reflects reduced inputs of new plant C

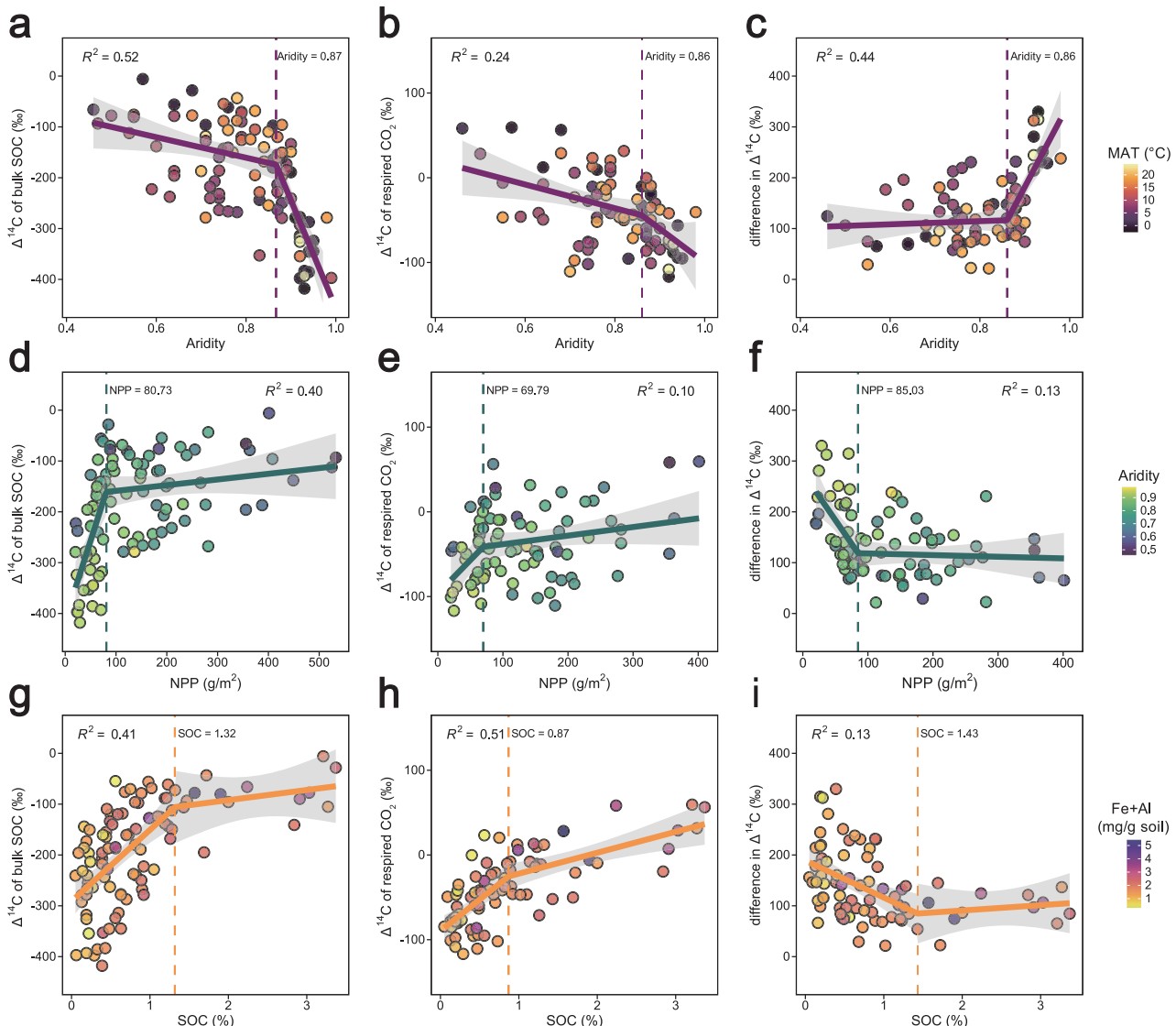

**Fig. 3 | Relationships between radiocarbon signatures and key abiotic and biotic drivers of soil carbon dynamics. a–c** $\Delta^{14}$C vs. aridity (1 − aridity index), colored by mean annual temperature (MAT). **d–f** $\Delta^{14}$C vs. net primary productivity (NPP), colored by aridity. **g–i** $\Delta^{14}$C vs. soil organic carbon (SOC), colored by oxalate-extractable Fe and Al (Fe + Al). $R^2$ indicates the proportion of variation explained by the segmented regression. Vertical dashed lines indicate thresholds from break-point analysis. Colored lines indicate regression fits, and shaded areas represent 95% confidence intervals.

(Fig. 3d) and preferential decomposition of recent C (Figs. 1b and 3b), leaving only a small fraction retained and stabilized over the long term (Fig. 3g). Notably, our results reveal an abrupt decline in bulk $\Delta^{14}$C when crossing an aridity threshold of c. 0.87 (Fig. 3a), exceeding the aridity thresholds observed for vegetation cover or productivity (as indicated by the Normalized Difference Vegetation Index, NDVI; aridity = 0.54) and SOC content (aridity = 0.70) in global drylands[42]. This suggests that the abrupt decline in SOC content at the 0.70 aridity threshold is driven by reduced contributions of recent C, while older C remains relatively less affected. However, when crossing the aridity threshold of 0.87, extreme reductions in plant C input (Fig. 3d), together with the increased age of C respired after soil rewetting (Fig. 3b), lead to a sudden decrease in $\Delta^{14}$C of remaining bulk SOC and the loss of millennia-old C (Fig. 3g).

The $\Delta^{14}$C of respired $CO_2$ was less sensitive to aridity and NPP but was better predicted by bulk SOC (Fig. 2b and Extended Data Fig. 9), without the pronounced SOC content threshold observed for bulk $\Delta^{14}$C (Fig. 3g and h). This is likely because respired $CO_2$ originates in part from bomb-derived recent C with relatively small $\Delta^{14}$C differences

across sites. SOC content could indicate a limitation of the amount of substrate available for decomposition, especially as soil incubations were conducted under moist conditions that might occur only intermittently in the field. Although SOC content correlated positively with oxalate-extractable Fe and Al (Extended Data Fig. 10), their concentrations showed no significant relationship with the $\Delta^{14}$C of bulk SOC (Fig. 2a and Extended Data Table 9). Similar patterns have been observed in African drylands, where the presence of poorly crystalline minerals does not necessarily correspond to older SOC ages[43]. This is likely due to the alkaline nature of the dryland soils, where Fe and Al oxides tend to carry net negative charges[44,45], thereby repelling negatively charged SOC and limiting the formation of mineral-associated organic C.

The paired bulk SOC and respired $\Delta^{14}$C data are an important feature of our study, as they emphasize differences in the controls of faster- and slower-cycling SOC and fill a critical data gap for radiocarbon in drylands. Although soils are known to contain C pools cycling on different timescales, radiocarbon-based estimates of SOC age are most often derived only from bulk $\Delta^{14}$C measurements,

combined with assumptions of homogeneity[14,19]. However, our results show that respired C is consistently younger than bulk SOC, with an increasing divergence between bulk and respired $\Delta^{14}C$ as aridity increases and NPP and SOC content decline (Figs. 2c and 3c, f, i). This suggests that radiocarbon-based age estimates from bulk SOC alone would substantially obscure the response of SOC on the decadal timescales associated with land management changes in drylands. Furthermore, our data also indicate that a portion of bulk SOC typically considered stable is susceptible to change on short timescales, particularly when dry soils are rewetted. Future $\Delta^{14}C$ measurements of chemically or physically defined fractions (e.g., POC and MAOC), as well as compound-specific analysis (e.g., lignin phenols, amino sugars, and black carbon), could further improve characterization of fast- and slow-cycling SOC pools. Together, these results underscore the importance of combining $\Delta^{14}C$ analysis of bulk SOC and respired $CO_2$ to derive more accurate estimates for constraining model simulations.

Our results have strong implications for understanding and predicting SOC persistence and transit time in global drylands under land management and climate change. We show that SOC is mostly derived from old C fixed centuries to millennia ago, with little contribution from bomb-derived recent C across global dryland sites. Our large-scale $\Delta^{14}C$ analysis of respired $CO_2$ reveals that despite preferential decomposition of relatively young C, even millennia-old C that is often assumed to be physically or chemically protected from decomposition can decompose after rewetting. Thus, the large soil $CO_2$ pulses observed after rainfall and the pronounced interannual variability of dryland C fluxes likely arise not only from recently assimilated C, as widely assumed, but also from the breakdown of centuries- to millennia-old C, revealing a previously underappreciated pathway for long-stored C to re-enter the atmosphere. The mean transit time for the microbially available pool of SOC derived from the $\Delta^{14}C$ of respired $CO_2$ (c. 520 years) is roughly 10 times longer than turnover time estimates from machine learning and ESMs, suggesting that dryland SOC cycling processes are not well represented in current models. In particular, ESMs that focus on contributions of recent plant C inputs and fail to account for the release of older C are likely to underestimate the magnitude and variability of dryland $CO_2$ fluxes.

In contrast to wetter ecosystems, aridity rather than temperature emerges as the dominant environmental control on SOC persistence and transit time in global drylands, largely through its effects on plant inputs and SOC content. As aridity intensifies with ongoing climate change[42,46], particularly beyond the identified threshold of 0.87, SOC persistence and transit time are expected to increase substantially. This increase, however, comes at the cost of a diminished capacity to store older C securely and to sequester new C from the atmosphere. While land management in drylands, such as afforestation, can reduce erosion and enhance plant inputs and soil C stocks, the associated increase in SOC cycling suggests that the capacity to store additional C will slow over time. Collectively, these changes could weaken the role of drylands as long-term C sinks and accelerate the return of long-stored C to the atmosphere, amplifying carbon–climate feedbacks. Recognizing these dynamics is crucial for improving ESM projections and informing land management strategies aimed at sustaining C storage in a warming and drying world.

## Methods
### Study sites
Soil samples were collected from 97 dryland ecosystems across six continents, along an aridity gradient (1 – aridity index, where aridity index is calculated as the mean annual precipitation divided by the mean annual potential evapotranspiration[47]) ranging from 0.46 to 0.99 (Extended Data Table 8). All sites are grouped into ten regions: Argentina, Australia, the alpine region of China, the Loess Plateau of China, the east region of China, the west region of China, Iran, South Africa, Spain, and the United States. The field sites cover a wide range

of vegetation types, including forest, grassland, shrubland, desert, and alpine meadow, with mean annual temperature (MAT) ranging from −3.4 to 24.0 °C and mean annual precipitation (MAP) from 28 to 754 mm.

### Sample collection
All topsoil sampling was conducted between June 2015 and July 2020. At each site in Argentina, Australia, Iran, South Africa, Spain, the United States, the China Loess Plateau region, the China east region, and the China west region[48,49], we established a 45 × 45 m² plot for collecting field data and samples. Within each plot, we randomly placed five quadrats, spaced at least 3 m apart, under the canopy of the dominant perennial vegetation, because these microsites accumulate litter, contain more nutrients, and support higher soil microbial activity[50,51]. Consequently, they best represented the biologically active surface SOC pool[52]. At the sites in Argentina, Australia, Iran, South Africa, Spain, and the United States, we followed the BIODESERT protocol[49] and sampled the 0–7.5 cm soil layer. In the Chinese regions, surface soil was sampled from 0 to 10 cm to be consistent with existing national survey[53]. Following the research guidelines during the second Tibetan Plateau Scientific Expedition, we established a 40 × 40 m² plot at each site in the alpine region of China. Within each plot, five 10 × 10 m² subplots were established, and three 1 × 1 m² survey quadrats were placed in each subplot for collecting topsoil samples (0–10 cm). Soil samples from all quadrats in each plot were combined to create one composite sample.

All sites had either not been grazed or had been only lightly grazed in the years prior to sampling. Following sampling, soil samples were transported to the laboratory in coolers as soon as possible to minimize soil decomposition[49]. Samples were then sieved (Ø 2 mm) to remove plant debris and rocks, air-dried at room temperature, and split into two subsamples: one was used for physico-chemical analyses, and the other was stored until incubation experiments in 2023.

### Soil incubation
Approximately 30 g of dry-weight soil was placed in 570 mL airtight flasks and adjusted to 60% of water-holding capacity (WHC). The flasks were immediately sealed and flushed with synthetic air (Rießner-Gase GmbH, Lichtenfels, Germany) to remove the $CO_2$ inside. To maintain humidity within the flasks during incubation, an open tube containing 5 mL of Milli-Q water was placed inside each flask[54]. Soil samples were incubated at 20 °C in a dark, temperature-controlled chamber. During the incubation period, respired $CO_2$ concentrations were measured by LI-COR 6262 (Lincoln, Nebraska, USA) on days 3, 7, 14, and 21. When the amount of respired C exceeded 1% of total soil C and surpassed 0.2 mg C in total[55], the headspace gas was collected and transferred for stable isotope measurement and extracted for radiocarbon analysis. Incubation of archived soil samples has proven to be a promising tool for radiocarbon research, as air-drying, rewetting, and storage time have been shown to have relatively small effects on the measured $\Delta^{14}C$ of respired $CO_2$[56].

### Stable isotope analyses
The $\delta^{13}C$ of total soil C was measured by an elemental analyzer (Carlo Erba 1100 CE analyzer; Thermo Fisher Scientific) coupled to an isotope ratio mass spectrometer (IRMS; Delta+ XL; Thermo Fisher Scientific) with a ConFlow III open-split (Finnigan MAT, Bremen, Germany)[57]. To measure the $\delta^{13}C$ of SOC, carbonates were removed from the soil samples by adding 5–6% $H_2SO_3$, and the $\delta^{13}C$ of SOC was then analyzed using the same IRMS procedure. The $\delta^{13}C$ of soil inorganic carbon (SIC) was calculated using the following equation:

$$\delta^{13}C_{TC} \times w_{TC} = \delta^{13}C_{SOC} \times w_{SOC} + \delta^{13}C_{SIC} \times w_{SIC} \qquad (1)$$

where $\delta^{13}C_{TC}$, $\delta^{13}C_{SOC}$, and $\delta^{13}C_{SIC}$ refer to the $\delta^{13}C$ values of total C, SOC, and SIC, respectively, and $w_{TC}$, $w_{SOC}$, and $w_{SIC}$ refer to the corresponding C contents.

To measure the $\delta^{13}C$ of $CO_2$, 12 mL of gas was collected from the incubation flasks using vacuum Labco extainers (Labco Ltd, Lampeter, UK). Gas samples were loaded via an autosampler (CTC Combi-PAL autosampler, CTC-Analytics, Zwingen, Switzerland), passed through a gas chromatograph (GC, Thermo Fisher Scientific, Bremen, Germany) connected to a ConFlow III open-split, and then transferred to the IRMS for analysis[58].

### Radiocarbon measurement and correction

Before radiocarbon ($^{14}C$) measurement, $CO_2$ from combusted bulk soils and respired $CO_2$ from incubations were purified and graphitized. Then the graphitized samples were analyzed using an accelerator mass spectrometer (AMS; Micadas, Ionplus, Switzerland) at the Radiocarbon Laboratory of MPI-BGC[59]. The $\Delta^{14}C_{sample}$ (per mil deviation in $^{14}C/^{12}C$ ratio from an absolute standard)[16] was corrected as follows:

$$\Delta^{14}C_{sample} = (F^{14}C \times e^{\lambda_c(1950-t)} - 1) \times 1000‰ \qquad (2)$$

where $F^{14}C$ is the Fraction Modern, defined as the ratio of the measured sample (normalized to a $\delta^{13}C$ value of $-25‰$) to 0.95 times the measured ratio of the Oxalic Acid I standard (OX-I), $\lambda_C$ refers to the updated radiocarbon decay constant (equals $1/8267$ yr$^{-1}$), and $t$ refers to the year of sampling.

To facilitate comparison across samples collected in different years (from 2015 to 2020), all reported $\Delta^{14}C$ values of our sites were background corrected based on the $\Delta^{14}C$ of the atmosphere in the year of sample collection ($\Delta^{14}C_{atmosphere}$)[17,60]:

$$\Delta^{14}C = \Delta^{14}C_{sample} - \Delta^{14}C_{atmosphere} \qquad (3)$$

where $\Delta^{14}C_{sample}$ is obtained from Eq. (2)[61].

### Estimation of the contribution of old SOC to respired $CO_2$

The one-pool model indicates that the mean age of respired C (i.e., transit time) from bulk soils averages $520 \pm 30$ years (Fig. 1a), ranging from a few years up to 1200 years. Because this is much younger than the mean age of bulk SOC ($2100 \pm 140$ years, Fig. 1a), the respired $CO_2$ must originate from at least two C pools cycling on different timescales, with a faster pool contributing most of the respired $CO_2$ and a slower pool dominating the bulk SOC age. We assumed that the fast pool contains recently fixed C, and we considered two cases for the $\Delta^{14}C$ signature of the older C: one is that the slow-cycling pool has the same $\Delta^{14}C$ as bulk SOC, and the other is that the bulk SOC age represents a mixture of modern C and $^{14}C$-free petrogenic C ($-1000‰$). The proportion of old C in soil respired $CO_2$ ($f_{old}$) was estimated by the following equations:

$$\Delta^{14}C_{CO_2} = \Delta^{14}C_{old} \times f_{old} + \Delta^{14}C_{young} \times f_{young} \qquad (4)$$

$$f_{old} + f_{young} = 1 \qquad (5)$$

where $\Delta^{14}C_{CO_2}$ represents the $\Delta^{14}C$ of total $CO_2$ released from the soil, and $\Delta^{14}C_{young}$ represents the $\Delta^{14}C$ of the atmosphere in the year of sample collection ($\Delta^{14}C_{atmosphere}$). If we assume $\Delta^{14}C_{old}$ equals the $\Delta^{14}C$ of bulk SOC, this provides the high-end estimate of the contribution of slow SOC to respiration. Under this assumption, the estimated mean $f_{old}$ is $23 \pm 3\%$ (mean $\pm$ SE), ranging from 6% to 69% across sites (Extended Data Table 4), even though this estimate involves uncertainties (e.g., the $\Delta^{14}C$ of the bomb-derived young C is most likely higher than that of atmospheric $CO_2$ in the sampling year). If we assume $\Delta^{14}C_{old}$ to represent the $^{14}C$-free petrogenic C, the estimated

$f_{old}$ (hereafter $f_{old, petro}$) is $4 \pm 0.4\%$, ranging from 1% to 12% (Extended Data Table 4). We consider the second assumption to be less likely than the first assumption (see "Results and Discussion"). If bulk SOC contains mostly intermediate-aged C (e.g., average of centennial to millennial ages), a more reasonable assumption is that $f_{old}$ is greater than 4%, but lower than 23%.

For the 41 sites from BIODESERT, we had data on POC and MAOC contents[12]; these fractions are often used to approximate fast and slow cycling C pools[31] and can help us make a better approximation of $f_{old}$. We assumed that POC (mostly assumed to be fast-cycling) has modern $\Delta^{14}C$ values, and estimated $\Delta^{14}C_{MAOC}$ using a mass balance:

$$bulk\ SOC\Delta^{14}C = \Delta^{14}C_{POC} \times w_{POC} + \Delta^{14}C_{MAOC} \times w_{MAOC} \qquad (6)$$

where $\Delta^{14}C_{POC}$ is assumed to reflect modern C ($\approx 0‰$), and $w_{POC}$ and $w_{MAOC}$ represent the fractions of POC and MAOC to SOC, respectively. Under these assumptions, $\Delta^{14}C_{MAOC}$ equals bulk $\Delta^{14}C/w_{MAOC}$. This estimated $\Delta^{14}C_{MAOC}$ was then used as the old endmember ($\Delta^{14}C_{old}$) in the two-endmember mixing model (Eqs. 4–5) to quantify the contribution of MAOC to respiration. This yields a mean $f_{old, MAOC}$ of $14 \pm 2\%$, with values ranging from 3 to 53% (Extended Data Table 4).

### Estimation of soil mean age and transit time

The mean age of bulk SOC and the mean transit time of soil respired C were estimated from $\Delta^{14}C_{sample}$ using two separate one-pool steady-state models: one fitted to the $\Delta^{14}C$ of bulk SOC and one fitted to the $\Delta^{14}C$ of respired $CO_2$. In this framework, bulk SOC is treated as a slow pool representing the majority of SOC mass, whereas respired $CO_2$ is derived from a fast pool representing a small fraction of total SOC that is most readily decomposed[16]. Each pool is represented by its own one-pool model and is assumed to be homogeneous and at steady state (i.e., not accumulating or losing C). The one-pool model that fitted to match the observed $\Delta^{14}C_{incubation}$ for fast and $\Delta^{14}C_{bulk}$ for slow was run from 1900 to the date of sampling (i.e., including inputs from C fixed during the time since nuclear weapons testing), to obtain the best fit to the observation[15,19]. For the one-pool model, the transit time, turnover time, and the mean age of C are all equal[60]. By performing separate calculations using the incubation (younger) and bulk soil (older), we approximate the dynamics of two homogeneous pools, one accessible to microbial metabolism and one representing the majority of soil organic C mass. In some cases, the $\Delta^{14}C_{sample}$ of respired $CO_2$ yielded two possible solutions (14 sites); for these, we selected the longer transit time[62], though selecting the younger ages would lead to a mean age of respired $CO_2$ of 520 years. All calculations were performed with the SoilR package in R[63].

Because the $\Delta^{14}C$ signatures of respired $CO_2$ during incubations differ from those of bulk SOC (see "Results and Discussion"), our assumption above that bulk SOC represents a homogeneous pool is at best an approximation. We therefore also applied two-pool models with different structures (two-pool parallel and two-pool series models; see Supplementary Text) to estimate the transit times of soil C[63–65]. These models use the FME[66] together with SoilR[63] R packages to identify parameters that best fit the observed $\Delta^{14}C_{bulk}$ and $\Delta^{14}C_{incubation}$ measurements in the year of sampling. Assuming steady state, the fitted parameters (decay rates in fast and slow pools and partitioning of inputs into each pool) allow calculation of the fraction of the mass of bulk soil C in each pool and their relative contributions to total respired $CO_2$. The two-pool models also allow estimating the transit time and age distributions[24]. Comparison of the results for mean transit time and system age for two-pool models to our estimate of the mean transit time and age from the one-pool model for the mean $\Delta^{14}C$ of bulk SOC and $\Delta^{14}C$ of respired $CO_2$ averaged across the arid and semi-arid sites respectively and show good agreement (Extended Data Table 7). From the two-pool model results, we also find that (1) more than 90% of the bulk C resides in the slow pool, and (2) depending on

the model, the slow pool contributes 9.9–54.5% of the C in respired $CO_2$, with the majority of respired C derived from relatively young sources (transit time of the fast pool averages) (Extended Data Table 7b). Given that the one-pool models provide reasonable approximations of fast and slow pool dynamics and considering the large uncertainties of fitting only two points in two-pool models, we report the one-pool model results in the main text.

### Uncertainty analysis of SIC contribution to CO₂ efflux ($f_{SIC}$)

We first assumed that carbonate was in equilibrium with $CO_2$ produced by SOC decomposition and treated all carbonate-derived $CO_2$ (solid SIC and its dissolution to DIC) as sharing a single carbonate end-member, and calculated the $\delta^{13}C$ of $CO_2$ in equilibrium with SIC (assuming a calcite–$CO_2$ fractionation of -9.6‰ at 20 °C)[67], denoted as $\delta^{13}C$-SIC$_{equilibrium}$ in Eq. (7):

$$\delta^{13}C - SIC_{equilibrium} = \delta^{13}C_{SIC} - 9.6‰ \quad (7)$$

where $\delta^{13}C_{SIC}$ represents the measured $\delta^{13}C$ of SIC in samples.

We then estimated the potential contribution of SIC to respired $CO_2$ ($f_{SIC}$) by a two-end-member mixing model as follows:

$$\delta^{13}C - CO_2 = f_{SIC} \times \delta^{13}C - SIC_{equilibrium} + f_{SOC} \times \delta^{13}C - SOC \quad (8)$$

$$f_{SIC} + f_{SOC} = 1 \quad (9)$$

where $\delta^{13}C$-$CO_2$ represents the $\delta^{13}C$ of the $CO_2$ released from the soil, $\delta^{13}C$-SIC$_{equilibrium}$ represents the $\delta^{13}C$ of $CO_2$ in equilibrium with SIC, $\delta^{13}C$-SOC represents the $\delta^{13}C$ of SOC-derived $CO_2$[68], and $f_{SOC}$ refers to the potential contribution of SOC to respired $CO_2$. The comparison between $\delta^{13}C$-$CO_2$ and $\delta^{13}C$-SOC is shown in Extended Data Fig. 4.

We finally estimated the $\Delta^{14}C$ of $CO_2$ derived from SOC ($\Delta^{14}C$-$CO_{2,\,SOC}$) using the following equation:

$$\Delta^{14}C - CO_2 = f_{SIC} \times \Delta^{14}C - CO_{2,SIC} + f_{SOC} \times \Delta^{14}C - CO_{2,SOC} \quad (10)$$

where $\Delta^{14}C$-$CO_2$ represents the $\Delta^{14}C$ of the $CO_2$ released from the soil, $\Delta^{14}C$-$CO_{2,\,SIC}$ and $\Delta^{14}C$-$CO_{2,\,SOC}$ refer to $\Delta^{14}C$ of SIC-derived $CO_2$ and SOC-derived $CO_2$, respectively, and $f_{SOC}$ refers to the potential contribution of SOC to respired $CO_2$.

Based on this estimation, we excluded 15 sites that exhibited high $f_{SIC}$ values (> 15% of respired $CO_2$ could reflect $\delta^{13}C$ derived from SIC), as these sites implied large differences between original $\Delta^{14}C$-$CO_2$ and the $f_{SIC}$-adjusted $\Delta^{14}C$-$CO_2$ (i.e., $\Delta^{14}C$-$CO_{2,\,SOC}$; Extended Data Fig. 5). These differences may reflect uncertainty in the $f_{SIC}$ adjustment because $\delta^{13}C$ of SOC-derived $CO_2$ may differ from bulk SOC when there are C4 plant-derived organic C inputs[69]. In addition, the assumption that SIC and SIC-derived $CO_2$ have the same $\delta^{13}C$ or $\Delta^{14}C$ values may not hold depending on which carbonate pool (e.g., DIC) equilibrates with respired $CO_2$, and we did not measure the speed at which isotopic equilibration occurs in our short-term incubations (subsequent experiments show this can take several days). Because four excluded sites were located on the Loess Plateau, we also removed the remaining two sites from the region to avoid insufficient representation along the gradient. For the remaining sites with $f_{SIC}$ values below 15%, we opted to use the original $\Delta^{14}C$ of respired $CO_2$ in our analysis, as SIC effects were minor and had no significant effect on our results (Extended Data Fig. 6). In total, 80 out of 97 sites were included in the final analysis of $\Delta^{14}C$ of respired $CO_2$ and the differences in $\Delta^{14}C$ of bulk SOC and respired $CO_2$ (Extended Data Fig. 5).

### Environmental and soil variables

Climate (aridity index and MAT), vegetation (NPP, plant cover, and species richness [Spec. richness]), and soil (SOC, pH, the sum of soil clay and silt [Clay + silt], the sum of oxalate-extractable Fe and Al oxides [Fe + Al], and microbial respiration rate [Resp. rate]) factors were used to explain soil $\Delta^{14}C$ signatures. Aridity index (precipitation/potential evapotranspiration)[1] was obtained from Global Aridity Index database[70]. To facilitate interpretation, we used aridity (calculated as 1 – aridity index) to represent the level of aridity[71]. MAT data based on the geographic coordinates were obtained from the WorldClim[72]. NPP data were downloaded from the Moderate Resolution Imaging Spectroradiometer (MODIS) C6, with a spatial resolution of $500 \times 500$ m$^2$ for 2000–2020[73]. Plant cover is the estimated total vegetation cover (%) per plot, and species richness is the number of plant species per plot; both are averaged to the site level[74].

Total soil carbon (TC) was measured using a varioMAX Cube elemental analyzer (Analysensysteme GmbH, Langenselbold, Germany) by dry combustion at 1100 °C, and SIC was measured after heating to 450 °C for 16 hours to remove organic C[75]. Soil pH was determined using a pH electrode (pH meter 538, WTW, Weilheim, Germany) in a 1:2.5 soil:water suspension. Soil texture (clay, silt, and sand) was measured by the pipette method[76]. Oxalate-extractable Fe and Al oxides (Fe + Al) were extracted using an oxalic acid-ammonium oxalate solution to represent poorly crystalline oxyhydroxides[77,78]. Soil respiration rate [Resp. rate] was calculated from the increase in headspace $CO_2$ over time, and converted to $CO_2$ production per unit soil mass using the ideal gas law, following standard static-incubation protocols:

$$Resp.\,rate = \frac{dC}{dt} \cdot \frac{P\,V\,M_{CO_2}}{R\,T\,m_{soil}} \quad (11)$$

where Resp. rate represents the mean soil respiration rate (µg $CO_2$ g soil$^{-1}$ day$^{-1}$); $dC/dt$ refers to the rate of increase (slope) in headspace $CO_2$ concentration over time (ppm day$^{-1}$); $P$ is the headspace air pressure (atm); $V$ is the headspace volume of the incubation vessel (L); $R$ is the ideal gas constant (0.082057 L atm mol$^{-1}$ K$^{-1}$); $T$ is the incubation temperature (K); $M_{CO_2}$ is the molar mass of $CO_2$ (44.01 g mol$^{-1}$); and $m_{soil}$ is the dry soil mass (g).

### Statistical analysis

We first assessed whether the $\Delta^{14}C$ values of bulk SOC ($n = 97$) and respired $CO_2$ ($n = 80$) differed significantly by a Mann–Whitney U test[79]. We then used linear mixed-effects regression models to assess the relative importance of climate, vegetation, and soil factors (independent variables) in explaining the $\Delta^{14}C$ of bulk SOC, respired $CO_2$, and their differences (response variables). All continuous predictors were standardized. Region was included as a random effect to account for variation among regions, allowing us to analyze the relationships between the $\Delta^{14}C$ variables and the key environmental and soil predictors within regions while estimating overall effects across all regions. Model assumptions were assessed using residual plots. NPP and soil respiration rate were natural-logarithm transformed to reduce skewness. Multicollinearity among predictors was checked by examining the variance inflation factor (VIF), with all VIF values below 3, indicating that collinearity was within an acceptable range and that the predictors could be considered sufficiently independent for modeling purposes[80] (Extended Data Table 12). Despite a significant negative relationship between NPP and aridity (Extended Data Fig. 11), both variables were included in the linear mixed-effects regression model as their VIF values were below 3 (Extended Data Table 12) and their interaction term was not significant (Extended Data Table 13). All analyses were performed using the lme4 R package[81].

To investigate whether $\Delta^{14}C$ values exhibit abrupt shifts with changing environmental and soil conditions, we performed piecewise linear regression analyses to explore the relationships between the $\Delta^{14}C$ values of bulk SOC and respired $CO_2$ (and their differences), and selected environmental variables. The segmented R package was used

to identify potential responses and ecological thresholds[82], which allows the estimation of breakpoints (i.e., thresholds) in the relationships by iteratively fitting linear regressions with changing slopes. To evaluate the legitimacy of introducing thresholds, we compared the Akaike Information Criterion (AIC) values of the piecewise regression models to those of simple linear models (Extended Data Table 14). Once a breakpoint was identified, separate linear models were fitted to the data segments on either side of the threshold. The identified threshold values represent points at which the response variables (the $\Delta^{14}C$ values of bulk SOC, respired $CO_2$, and their differences) exhibit a significant shift in response to changes in the predictor variables (aridity, NPP, and SOC content).

We also used a random forest model to test potential nonlinear relationships and to predict how $\Delta^{14}C$ dynamics may respond to environmental changes (Extended Data Fig. 12). The random forest approach is known to mitigate overfitting in the training dataset and address problems related to multicollinearity[83]. We conducted a 10-fold cross-validation to ensure that the $\Delta^{14}C$ value from each site was entirely included in either the 70% training dataset or 30% test dataset for model validation. The root mean square error (RMSE) was calculated on the testing dataset to assess model performance. Permutation feature importance was applied to determine the importance of each independent variable. Additionally, partial dependence plots (PDP) were used to illustrate the influence of each important explanatory variable (aridity, NPP, and SOC content) on the predicted outcomes of the random forest model, while accounting for the influence of all other predictors[84] (Extended Data Fig. 13). Relevant analyses were performed using the mlr3[85] and iml R packages[86].

## Data availability
The raw data generated in this study are deposited in figshare (https://doi.org/10.6084/m9.figshare.30256888).

## Code availability
The R code used for the analyses in this study is available on Zenodo (https://doi.org/10.5281/zenodo.18788036).

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

## Acknowledgements

Funding: This research was jointly supported by the Max Planck Society grant M.FE.A.EBIO0002 (J.H., S.T., S.Z.) and Chinese Academy of Sciences grant HZXM20225001 (N.L., B.F., Y.Z.), European Research Council grant 647038 (BIODESERT) (F.T.M.), and King Abdullah University of Science and Technology (F.T.M. and E.G.). We thank all the participants in the BIODESERT global survey for conducting fieldwork. We also thank Iris Kuhlmann, Ines Hilke, Heiko Moossen, Heike Geilmann, Petra Linke, Michael Raessler, Jeffrey Beem-Miller, and Theresa Klötzing, and the radiocarbon team of Max Planck Institute for Biogeochemistry for their assistance with laboratory work. We thank Ulrich Weber and Yuchen Bai for help with data extraction.

## Author contributions

Conceptualization: J.H., S.T., N.L., and B.F.; Methodology: H.W., J.H., S.T., F.T.M., N.L., C.W., G.Z., W.C., M.S., S.F.F., D.S.E.B., AT.E., and K.W.; Investigation: H.W., J.H., S.T., N.L., C.A.S., G.Z., V.O., K.W., W.C., D.J.E., N.G., Y.L.B.-P., H.S., B.G., S.A., C.P., E.G., M.G.G., E.V., J.J.G., J.M.V., Y.W., and B.J.M.; Visualization: H.W.; Funding acquisition: J.H., S.T., S.Z., N.L., B.F., and Y.Z.; Project administration: J.H., S.T., S.Z., N.L., B.F., and Y.Z.; Supervision: J.H., S.T., N.L., B.F., S.Z., and M.A.D.; Writing—original draft: H.W., J.H., and S.T.; Writing—review and editing: J.H., S.T., F.T.M., N.L., M.A.D., S.Z., B.F., G.Z., Y.Z., C.A.S., M.S., D.S.E.B., S.F.F., A.T.E., K.W., C.W., C.P., D.J.E., E.G., H.S., E.V., J.M.V., and N.G.

## Funding

## Competing interests

The authors declare no competing interests.

## Additional information

¹Department of Biogeochemical Processes, Max Planck Institute for Biogeochemistry, Jena, Germany. ²Geo-Biosphere Interactions, Department of Geosciences, University of Tübingen, Tübingen, Germany. ³Environmental Science and Engineering, Biological and Environmental Science and Engineering Division, King Abdullah University of Science and Technology, Thuwal, Saudi Arabia. ⁴State Key Laboratory of Regional and Urban Ecology, Research Center for Eco-Environmental Sciences, Chinese Academy of Sciences, Beijing, China. ⁵National Observation and Research Station of Earth Critical Zone on the Loess Plateau in Shaanxi, Xi'an, China. ⁶Shaanxi Yan'an Forest Ecosystem Observation and Research Station, Beijing, China. ⁷Lhasa Plateau Ecosystem Research Station, Key Laboratory of Ecosystem Network Observation and Modeling, Institute of Geographic Sciences and Natural Resources Research, Chinese Academy of Sciences, Beijing, China. ⁸College of Resources and Environment, University of Chinese Academy of Science, Beijing, China. ⁹Instituto Multidisciplinar para el Estudio del Medio Ramón Margalef, Universidad de Alicante, Alicante, Spain. ¹⁰Fujian Academy of Forestry, Fuzhou, China. ¹¹Centre for Ecosystem Science, School of Biological, Earth and Environmental Sciences, University of New South Wales, Sydney, NSW, Australia. ¹²Instituto de Ciencias Agrarias (ICA), CSIC, Madrid, Spain. ¹³Departamento de Ingeniería y Morfología del Terreno, Escuela Técnica Superior de Ingenieros de Caminos, Canales y Puertos, Universidad Politécnica de Madrid, Madrid, Spain. ¹⁴Université Clermont Auvergne, INRAE, VetAgro Sup, Unité Mixte de Recherche Ecosystème Prairial, Clermont-Ferrand, France. ¹⁵IMBE, Aix Marseille Univ, CNRS, Avignon Université, IRD, Aix-en-Provence, France. ¹⁶Estación Experimental de Zonas Áridas (EEZA), CSIC, Almería, Spain. ¹⁷Departamento de Biología y Geología, Física y Química Inorgánica, Universidad Rey Juan Carlos, Madrid, Spain. ¹⁸Instituto Universitario de Investigación en Olivar y Aceite de Oliva-INUO, Universidad de Jaén, Jaén, Spain. ¹⁹Department of Agricultural and Food Chemistry, Faculty of Sciences, Universidad Autónoma de Madrid, Madrid, Spain. ²⁰Departamento de Ciencias Agrarias y Medio Natural, Escuela Politécnica Superior, Instituto Universitario de Investigación en Ciencias Ambientales de Aragón (IUCA), Universidad de Zaragoza, Huesca, Spain. ²¹Departamento de Biodiversidad, Ecología y Evolución, Facultad de Ciencias Biológicas, Universidad Complutense de Madrid, Madrid, Spain. ²²Department of Land Resources and Environmental Sciences, Montana State University, Bozeman, MT, USA. ²³State Key Laboratory of Loess Science, Institute of Earth Environment, Chinese Academy of Sciences, Xi'an, China. ²⁴Department of Biogeochemical Signals, Max Planck Institute for Biogeochemistry, Jena, Germany. ✉e-mail: nanlv@rcees.ac.cn; hjianbei@bgc-jena.mpg.de

