## [Peer Review file · Nature Communications]

Persistence and turnover of soil organic carbon in global drylands

Corresponding Author: Dr Jianbei Huang

Version 0:

Reviewer comments:

Reviewer #1

(Remarks to the Author)

Summary

This study provides a novel and valuable global dataset of radiocarbon measurements from topsoil across 97 dryland sites. It challenges the long-standing assumption that old SOC in dryland ecosystems is largely unavailable for decomposition. The main findings are: (1) fast-cycling carbon pools (respired CO₂) have turnover times orders of magnitude longer than previously estimated by ESMs and incubation experiments, and (2) aridity (rather than temperature), net primary productivity (NPP), and SOC content are the dominant predictors of both bulk SOC and respired CO₂ ages.

Overall, the dataset is unique and provides strong empirical evidence that current ESMs and assumptions substantially underestimate the turnover time of fast-cycling carbon pools. The manuscript would benefit from a major revision to allow clearer preparation of the main and supplementary figures, as several features referenced in the text are missing or difficult to locate (see detailed comments below). I also recommend revising the abstract and introduction to make the narrative more accessible to a broader audience.

Major comments

- I find the structure of the Results section confusing. Most of the results presented from the beginning of the section up to the section discussing “Predictors of radiocarbon signatures in global dryland soils” rely on data shown in the Extended Data, while Fig. 2 does not appear to support many of the claims discussed. For instance, I cannot determine how the authors derived the value 23% in Line 152 from any main-text or supplementary figure, except for a similar statement in the Method. I strongly recommend reorganizing/re-visiting the figures and reassessing which results should be presented as main findings.
- Regarding Line 493, are the aridity index and NPP truly independent? Higher aridity typically reduces vegetation productivity, so these variables are expected to be strongly correlated. I recommend adding a scatterplot of aridity index versus NPP to clarify their relationship.

Minor comments

- I recommend that the authors briefly explain the terms bulk SOC and respired CO₂ or use more general phrasing, especially in the abstract, so the study is accessible to readers from other fields.
- Line 159: These appear to be the main results—why are they placed in the Supplementary Information?
- Lines 175–178: Please provide references to support this statement.
- Line 204: The finding is not necessarily in “contrast” to ref. 18. That reference includes more sites across diverse ecosystems, not just drylands. Therefore, different feature importance rankings are expected. “Contrast” may not be the appropriate term here.

Fig. 1

- Fig. 1 is a bit busy; I suggest removing all the zoom-in circles. In addition, most sites fall within high-aridity regions, and since the aridity gradient adds little additional information, the authors can consider removing it.
- The authors can also consider moving Figure 1 to the supplementary section since Figure 1 does not present new findings, and only shows the distribution of sites measured in this study.

Fig. 2

- Line 140: Use “relative magnitude” instead of “magnitude.”
- The purple and orange colors are not mentioned in the legend and do not convey additional information. I recommend removing them.
- Lines 137–138: The phrase “generally exceeds” requires statistical support. Please include the relevant statistical test and indicate significance directly on the boxplot.

-
Fig. 3

- Please explain the uncertain bars in the figure.

Method

- Lines 303–314: In the sample collection section, please explain why samples were collected under canopy cover at certain locations and why a different sampling procedure was used in the China regions.
- Line 317: Specify the storage conditions (temperature, duration) and discuss whether transportation may have affected sample quality.
- I may have overlooked it, but I could not find a description of how soil respiration rate [Resp. rate] was measured (Line 467). Please add this to the Methods.

Supplementary Files

- Extended Data Fig. 3: I did not see Aridity Index in the list of features. The authors should also check the statement at Line 203.

Reviewer #2

(Remarks to the Author)

This manuscript details a novel study on the global dryland soil organic carbon persistence and turnover. This manuscript fills in a very important, yet missing piece of our understanding of global soil organic carbon persistence in these understudied ecosystems. This manuscript clearly articulates the major gap in drylands research thus far and adds an impressive number of sites and global coverage.

This manuscript should be published with some corrections I have described below. My comments mostly focus on some portions of the text that can be clarified. Some discussion of inorganic contribution to the CO₂ incubation data which needs to be mentioned in the main text (there is significant discussion in the material and methods, however, it was left as a question in my mind the entire time).

Also, in the drylands, is there a depth component to the soils? How deep are the soils and could there be a significant contribution of older or fossil carbon in these ecosystems because there has not been the flux of water to significantly break down this organic flux from the bedrock. If the soils are quite thin and vegetation is not contributing to the OC pool, could fossil carbon contribute more significantly thus driving OC ages older, but without contributing to SOC storage from the terrestrial environment or inputs? There are new papers discussing this source and drylands might be important when considering this flux of carbon into the ecosystems (Evans et al, 2025, Grant et al, 2023).

Additionally, one part of the discussion that is missing is where is the older portion of SOC coming from, you calculated that 23% of the CO₂ respired comes from older pools. Is there some thought on whether this is somehow from mineral associated OC that can be broken down or is it from POC that is not mineral associated? In arid regions where could this older pool that is vulnerable to respiration and degradation likely stored until it is then destabilized?

Specific line-by line comments

75: Is there an accepted definition of drylands? A certain metric of aridity or precipitation? I think that would be very useful in the first paragraph for those who are not ecosystem scientists, it puts the actual metrics (rainfall or aridity in perspective). I believe this is defined in the methods section, however, it is useful to mention it up front, so the reader can put it into context.

110: This is a subtle difference, but it is 97 dryland sites correct? Or are there 97 unique dryland ecosystems?

115: I find the wording of these hypotheses confusing. Could it be stated that reduced vegetation inputs drive older ages of SOC because there is less new OC incorporated into the soil? But wouldn't that be an effect of stronger mineral protection of the SOC that has already been stabilized? I would put the “with increasing aridity” in the first part of the sentence hypothesis. The reduced aridity is driving the reduction in vegetation inputs, correct?

117: This next sentence of the hypothesis seems to say that with increased aridity you would expect the respired CO₂ to become even younger because of a preferential decomposition of recent C. It seems to me to contradict the next sentence, or maybe I am missing something, but could it be more clear if there was an added statement of the 14C of the evolved CO₂ from the respired CO₂ would in actuality decrease in amount and age as the fresh inputs are rapidly consumed, and the only SOC able for decomposition must come from other, older sources?

139: Cite the aridity index calculation. Is it in the text or SI?

153: Could this be an artifact of the incubation? That the CO₂ evolved is somehow would not be that young in the field? Is there any discussion that in incubations there tends to be a preferential utilization of OC that was disturbed?

156: What about DIC in arid soils? It seems to me that could contribute significant older carbon either from parent material carbonates or pedogenic carbonates. This hasn't been mentioned yet in the text, and I would think in arid regions where there hasn't been much weathering the soils would be full of base cations that could form carbonates and contribute to the CO₂ flux when re-wetted.

157-161: This was the first mention of depth in this study. Were what depths were the samples taken from or used in this study? Was it all topsoil? And what is the topsoil definition that was being used here? Can that be directly compared to the other topsoil values? And how would you expect this to change with depth?

167: This is not a modeled value correct? Or is it a 1-pool value? Can you clarify.

175: Do you mean the estimates for only drylands broken out from the ESM models or drylands compared to entire biomes from the ESM model?

202: As mentioned before, can you briefly explain/define the aridity index?

258: I would argue other regions as well. What about the combination of comparing other fractions in these regions to ^{14}C incubations or different compound classes in dryland ecosystems. Would or should there be an effort to attempt measure different compounds that could accurately assess the fast-cycling pool of soil carbon, without some of the draw backs of incubations?

383: This wording is confusing. It says there is a one-pool steady state model, but then the next sentence explains this assumes there are two-pool represented by what you measured? Can you clarify this sentence.

462: I do think some of this discussion of soil inorganic carbon needs to be in the main text

Reviewer #1 (Remarks to the Author):

[Comment 1]

Summary

This study provides a novel and valuable global dataset of radiocarbon measurements from topsoil across 97 dryland sites. It challenges the long-standing assumption that old SOC in dryland ecosystems is largely unavailable for decomposition. The main findings are: (1) fast-cycling carbon pools (respired CO₂) have turnover times orders of magnitude longer than previously estimated by ESMs and incubation experiments, and (2) aridity (rather than temperature), net primary productivity (NPP), and SOC content are the dominant predictors of both bulk SOC and respired CO₂ ages.

Overall, the dataset is unique and provides strong empirical evidence that current ESMs and assumptions substantially underestimate the turnover time of fast-cycling carbon pools. The manuscript would benefit from a major revision to allow clearer preparation of the main and supplementary figures, as several features referenced in the text are missing or difficult to locate (see detailed comments below). I also recommend revising the abstract and introduction to make the narrative more accessible to a broader audience.

[Response] Thanks for your positive assessment and constructive comments. We have revised the figures and ensured that all important features are now clearly referenced in the text. We have also reviewed the abstract and introduction to improve the readability for a broader audience.

Major comments

[Comment 2] I find the structure of the Results section confusing. Most of the results presented from the beginning of the section up to the section discussing ‘Predictors of radiocarbon signatures in global dryland soils’ rely on data shown in the Extended Data, while Fig. 2 does not appear to support many of the claims discussed. For instance, I cannot determine how the authors derived the value 23% in Line 152 from any main-text or supplementary figure, except for a similar statement in the Method. I strongly recommend reorganizing/re-visiting the figures and reassessing which results should be presented as main findings.

[Response] We thank the reviewer’s constructive comment on the structure of Results section. We agree that the mean ages of bulk SOC and respired CO₂ derived from $\Delta^{14}\text{C}$ values are main results but not presented in the main figures in the original version. Thus, in the revised manuscript we have added a secondary y-axis (Mean age) on the right to Fig. 1a (see revised figure below), reporting the mean ages of bulk SOC and respired CO₂ together with the corresponding $\Delta^{14}\text{C}$ values. This allows the reader to link mean ages to radiocarbon signatures.

Fig. 1 | Radiocarbon signatures of global dryland soils. $\Delta^{14}\text{C}$ (‰) represents the deviation of a sample’s radiocarbon content from the preindustrial atmosphere (0‰). Positive values ($\Delta^{14}\text{C} > 0\text{‰}$) indicate the presence of ‘bomb ^{14}C ’ fixed in the past c. 60 years from atmospheric weapons testing, while negative values ($\Delta^{14}\text{C} < 0\text{‰}$) indicate C old enough for substantial radioactive decay (^{14}C half-life = 5,730 years). (a) $\Delta^{14}\text{C}$ of respired CO_2 significantly exceeds that of bulk SOC, with a corresponding mean ^{14}C age of 520 years compared to 2,100 years for bulk SOC. Mean ages were calculated from the $\Delta^{14}\text{C}$ values using one-pool models. The $\Delta^{14}\text{C}$ values and mean age associated with ‘bulk SOC’ are in purple, while the $\Delta^{14}\text{C}$ values and mean age associated with ‘respired CO_2 ’ are in orange.

We have also added a new table in the supplementary (Extended Data Table 3) to show the age of bulk SOC and respired CO_2 for each site and have presented the data used for f_{old} calculations in Extended Data Table 6.

[Comment 3] Regarding Line 493, are the aridity index and NPP truly independent? Higher aridity typically reduces vegetation productivity, so these variables are expected to be strongly correlated. I recommend adding a scatterplot of aridity index versus NPP to clarify their relationship.

[Response] Thanks for this insightful comment. In our database, it is reasonable to treat aridity and NPP as independent factors in the linear mixed regression model because all variance inflation factor (VIF) values were < 3 (indicating no collinearity). To clearly show this, we have now added the VIF values in Extended Data Table 12. In addition, we tested interactive effects among Aridity, NPP, and SOC and found that the p values of the Aridity \times NPP interaction on $\Delta^{14}\text{C}$ values were both > 0.05 (Extended Data Table 13), indicating that there was no significant interaction between aridity and NPP in the linear mixed model.

To examine the relationship between NPP and aridity, we plotted a scatterplot of aridity degree (1 – aridity index) versus NPP (Extended Data Fig. 11). We acknowledge that, although aridity and NPP are statistically treated as independent predictors of $\Delta^{14}\text{C}$ in the linear mixed effects models, they are ecologically linked, as high aridity is often associated with low vegetation productivity (Maestre et al. 2016, Berdugo et al. 2020).

Accordingly, we have revised the main text at lines 534–541: ‘Multicollinearity among predictors was checked by examining the variance inflation factor (VIF), with all VIF values below 3, indicating that collinearity was within an acceptable range and that the predictors could be considered sufficiently independent for modeling purposes⁸⁰ (Extended Data Table 12). Despite a significant negative relationship between NPP and aridity (Extended Data Fig. 11), both variables were included in the linear mixed-effects regression model as their VIF values were below 3 (Extended Data Table 12) and their interaction term was not significant (Extended Data Table 13).’

Extended Data Fig. 11 | Relationship between NPP and aridity across regions. The significant negative relationship between NPP and aridity ($R^2 = 0.43$) suggested water availability limits vegetation productivity in drylands. Significance levels: *** $p < 0.001$.

Minor comments

[Comment 4] I recommend that the authors briefly explain the terms bulk SOC and respired CO_2 ; or use more general phrasing, especially in the abstract, so the study is accessible to readers from other fields.

[Response] We agree with the reviewer that briefly explaining the terms ‘bulk SOC’ and ‘respired CO_2 ’ will help broader readers understand the study. ‘Bulk SOC’ refers to the total organic carbon stored in soil, and ‘respired CO_2 ’ refers to the CO_2 released by soil respiration during incubations. Given the 150 words limit in the abstract, we

revised lines 67–68 to read: ‘We measured radiocarbon in SOC and in CO₂ released from soil respiration from 97 dryland sites across six continents.’

[Comment 5] Line 159: These appear to be the main results — why are they placed in the Supplementary Information?

[Response] Thanks for the helpful comment. We agree that the mean age of bulk SOC and the mean age of respired C in drylands are the key results of our study and should be more visible, so we added a secondary y-axis to Fig. 1a, reporting the mean ages of bulk SOC and respired CO₂ together with the corresponding $\Delta^{14}\text{C}$ values. We also added a new Extended Data Table 3 in the supplementary to show the age of bulk SOC for each site (see our responses to your previous comment 2).

[Comment 6] Lines 175–178: Please provide references to support this statement.

[Response] Thanks for the reviewer’s helpful comment. We have added two references to support this statement at line 194 in the revised manuscript:

1. Zhang, L. *et al.* Mapping global distributions, environmental controls, and uncertainties of apparent topsoil and subsoil organic carbon turnover times. *Earth Syst. Sci. Data* **17**, 2605-2623 (2025).
2. Xiao, L. J. *et al.* Younger carbon dominates global soil carbon efflux. *Glob. Change Biol.* **28**, 5587-5599 (2022).

[Comment 7] Line 204: The finding is not necessarily in ‘contrast’ to ref. 18. That reference includes more sites across diverse ecosystems, not just drylands. Therefore, different feature importance rankings are expected. ‘Contrast’ may not be the appropriate term here.

[Response] Thanks for the useful suggestion. We have replaced ‘contrasts with’ by ‘differs from’ in lines 201–204: ‘This result differs from previous global syntheses that identified MAT as the dominant climatic factor, likely because those studies included a wider range of ecosystems but limited dryland data¹⁹’.

Figures

[Comment 8]

Fig. 1

- Fig. 1 is a bit busy; I suggest removing all the zoom-in circles. In addition, most sites fall within high-aridity regions, and since the aridity gradient adds little additional information, the authors can consider removing it.

- The authors can also consider moving Figure 1 to the supplementary section since Figure 1 does not present new findings, and only shows the distribution of sites measured in this study.

[Response] We thank the reviewer for the helpful suggestions regarding the visualization of Fig. 1. We have removed all zoom-in circles to make it clearer and used hollow circles to highlight the ten regions. We have moved Fig. 1 to the Supplementary section.

Extended Data Fig. 1 | Locations of the dryland sites for paired $\Delta^{14}\text{C}$ of bulk SOC and respired CO_2 . Dryland areas are characterized by aridity levels (1 – aridity index) higher than 0.35, where aridity index is calculated as the ratio of mean annual precipitation to mean annual potential evapotranspiration^{17,18}. The background of the global map represents the degree of aridity, with higher values indicating more arid regions. The hollow circles indicate samples collected from ten different regions worldwide. Further details about the surveyed field sites are provided in Extended Data Table 1.

[Comment 9]

Fig. 2

- Line 140: Use ‘relative magnitude’ instead of ‘magnitude’.

[Response] Thanks for the detailed comment. We have revised the text to use ‘relative magnitude’ (line 586).

[Comment 10] - The purple and orange colors are not mentioned in the legend and do not convey additional information. I recommend removing them.

[Response] Thank the reviewer for this helpful suggestion. We agree that the colors should be clearly defined in the legend. We prefer to keep the colors to maintain visual clarity. We have updated the legend (lines 581–583) to define the colors explicitly: The

$\Delta^{14}\text{C}$ values and mean age associated with ‘bulk SOC’ are in purple, while the $\Delta^{14}\text{C}$ values and mean age associated with ‘respired CO_2 ’ are in orange.

[Comment 11] - Lines 137-138: The phrase ‘generally exceeds’ requires statistical support. Please include the relevant statistical test and indicate significance directly on the boxplot.

[Response] Thanks for the helpful suggestion. Because $\Delta^{14}\text{C}$ of bulk SOC was not normally distributed, we used a Mann–Whitney U test to compare $\Delta^{14}\text{C}$ of bulk SOC ($n = 97$) and $\Delta^{14}\text{C}$ of respired CO_2 ($n = 80$) as two independent samples. The result showed that $\Delta^{14}\text{C}$ of respired CO_2 was significantly higher than that of bulk SOC ($W = 566$, $p < 0.001$). The significance level has also been indicated directly on the boxplot in Fig. 1a. In addition, we performed a paired analysis for the 80 sites where both $\Delta^{14}\text{C}$ of respired CO_2 and bulk SOC were available. Because the paired $\Delta^{14}\text{C}$ differences did not follow a normal distribution, we used a Wilcoxon signed-rank test. This analysis again confirmed that $\Delta^{14}\text{C}$ of respired CO_2 was significantly higher than that of bulk SOC ($V = 3,240$, $p < 0.001$).

Accordingly, we added the significance on the Fig. 1a and revised the lines 578–580 as: ‘(a) $\Delta^{14}\text{C}$ of respired CO_2 significantly exceeds that of bulk SOC, with a corresponding mean ^{14}C age of 520 years compared to 2,100 years for bulk SOC.’

In addition, we added the statistical test method in lines 524–525: ‘We first assessed whether the $\Delta^{14}\text{C}$ values of bulk SOC ($n = 97$) and respired CO_2 ($n = 80$) differed significantly by a Mann–Whitney U test⁷⁹.’

[Comment 12]

Fig. 3

- Please explain the uncertain bars in the figure.

[Response] Thank you for the useful suggestion. We have clarified this in the figure caption (line 597) as follows: ‘Error bars show 95% confidence intervals (CIs) of fixed-effect coefficients.’

Method

[Comment 13] Lines 303-314: In the sample collection section, please explain why samples were collected under canopy cover at certain locations and why a different sampling procedure was used in the China regions.

[Response] We thank the reviewer for this comment and agree that the rationale of our sampling design should be better explained. Dryland landscapes typically form a patchy mosaic of vegetated ‘islands’ and bare ground (Maestre et al. 2012, Maestre et al. 2021).

To standardize sampling across sites and target the biologically active “fertile-island” microsites shaped by plant-derived inputs and associated microbial activity, we collected soils from beneath the plant canopy. These soils accumulate more litter and roots, have higher microbial activity, and contain more SOC than bare soils (Schlesinger and Pilmanis 1998, Maestre et al. 2022a, Eldridge et al. 2024). However, the plot design in the alpine region of China differed from other regions because it was necessary to follow the research guidelines during the second Tibetan Plateau Scientific Expedition to minimize disturbance to fragile habitats (Sun et al. 2013).

We revised the sample collection part in the Methods (lines 310–319): ‘Within each plot, we randomly placed five quadrats, spaced at least 3 m apart, under the canopy of the dominant perennial vegetation, because these microsites accumulate litter contain more nutrients and support higher soil microbial activity^{50,51}. Consequently, they best represented the biologically active surface SOC pool⁵². At the sites in Argentina, Australia, Iran, South Africa, Spain, and the United States, we followed the BIODESERT protocol⁴⁹ and sampled the 0–7.5 cm soil layer. In the Chinese regions, surface soil was sampled from 0–10 cm to be consistent with existing national survey⁵³. Following the research guidelines during the second Tibetan Plateau Scientific Expedition, we established a 40 m × 40 m plot at each site in the alpine region of China.’

[Comment 14] Line 317: Specify the storage conditions (temperature, duration) and discuss whether transportation may have affected sample quality.

[Response] Thank the reviewer for this comment regarding storage conditions. All samples followed the standardized BIODESERT dryland protocol (Maestre et al. 2022a, Maestre et al. 2022b). At each site, soils were homogenized in the field and transported to the laboratory with coolers as soon as possible to minimize microbial respiration and SOC decomposition. Samples were then sieved, air-dried at room temperature, and stored until the incubation experiments in 2023. Beem-Miller et al. showed that storage duration had small effects on $\Delta^{14}\text{C}$ of respired CO_2 (Beem-Miller et al. 2021), supporting the use of air-dried archived soils for incubations.

Accordingly, we have revised lines 324–328 as follows: ‘Following sampling, soil samples were transported to the laboratory in coolers as soon as possible to minimized SOC decomposition⁴⁹. Samples were then sieved ($\text{\O} 2 \text{ mm}$) to remove plant debris and rocks, air-dried at room temperature, and split into two subsamples: one was used for physicochemical analyses, and the other was stored until incubation experiments in 2023.’

[Comment 15] I may have overlooked it, but I could not find a description of how soil respiration rate [Resp. rate] was measured (Line 467). Please add this to the Methods.

[Response] We thank the reviewer for this comment. During incubations, headspace CO_2 concentrations (ppm) were measured by LI-COR 6262 (Lincoln, Nebraska, USA) on day 3, 7, 14, 21. Soil respiration rate [Resp. rate; $\mu\text{g CO}_2/\text{g soil/day}$] (see Extended

Data Table 1) was calculated from the increase in headspace CO₂ over time, and converted to CO₂ production per unit soil mass using the ideal gas law, following standard static-incubation protocols:

$$\text{Resp. rate} = \frac{dC}{dt} \cdot \frac{P V M_{\text{CO}_2}}{R T m_{\text{soil}}}$$

where Resp. rate represents the mean soil respiration rate ($\mu\text{g CO}_2 \text{ g}^{-1} \text{ soil day}^{-1}$); dC/dt refers to the rate of increase (slope) in headspace CO₂ concentration over time (ppm day^{-1}); P is the headspace air pressure (atm); V is the headspace volume of the incubation vessel (L); R is the ideal gas constant ($0.082057 \text{ L}\cdot\text{atm}\cdot\text{mol}^{-1}\cdot\text{K}^{-1}$); T is the incubation temperature (K); M_{CO_2} is the molar mass of CO₂ (44.01 g mol^{-1}); and m_{soil} is the dry soil mass (g).

We have now added the details on how soil respiration rate [Resp. rate] was calculated in lines 513–522.

Supplementary Files

[Comment 16] Extended Data Fig. 3: I did not see Aridity Index in the list of features. The authors should also check the statement at Line 203.

[Response] Thanks for this helpful comment. We have removed Extended Data Fig. 3, as it was not appropriate in the context. We also revised the statement in lines 199–202 to read: ‘Aridity ($1 - [\text{precipitation}/\text{potential evapotranspiration}]$) was more important than MAT (Extended Data Table 9) in predicting variations in bulk $\Delta^{14}\text{C}$ in global drylands (Fig. 2a).’

Reviewer #2 (Remarks to the Author):

[Comment 1] This manuscript details a novel study on the global dryland soil organic carbon persistence and turnover. This manuscript fills in a very important, yet missing piece of our understanding of global soil organic carbon persistence in these understudied ecosystems. This manuscript clearly articulates the major gap in drylands research thus far and adds an impressive number of sites and global coverage.

[Response] We thank the reviewer for this positive and encouraging assessment of our work. We have revised the manuscript according to the specific comments provided below and we believe it is now clearer and stronger.

[Comment 2] This manuscript should be published with some corrections I have described below. My comments mostly focus on some portions of the text that can be clarified. Some discussion of inorganic contribution to the CO₂ incubation data which needs to be mentioned in the main text (there is significant discussion in the material and methods, however, it was left as a question in my mind the entire time).

[Response] Thanks for the reviewer's helpful comment. We agree that it is important to address potential contributions of SIC to respired CO₂ in the main text. We have added the following sentences in the main text (lines 155–169): 'When interpreting the $\Delta^{14}\text{C}$ of SOC-derived respired CO₂, we accounted for potential contributions from ¹⁴C-depleted SIC to the respired CO₂ that could bias apparent respiration ages^{34,35}. Rewetting soils can promote carbonate dissolution and dissolved inorganic carbon (DIC) formation, resulting in CO₂ contributions from both parent-material and pedogenic carbonates^{36,37}. For 45 of our sites, SIC content was below 0.1%, and we assumed all respired CO₂ was derived from SOC. For soils with SIC content above 0.1%, the potential SIC contribution to respired CO₂ (f_{SIC}) was estimated using a $\delta^{13}\text{C}$ -based two-end-member mass balance due to the distinct $\delta^{13}\text{C}$ signatures of SIC and SOC (see Methods; Extended Data Fig. 4). This estimate was then incorporated into a $\Delta^{14}\text{C}$ -based mass balance to estimate the influence of SIC on the $\Delta^{14}\text{C}$ of SOC-derived, respired CO₂. This resulted in the exclusion of 17 sites with high f_{SIC} (mostly > 15%) to avoid substantial underestimation of the $\Delta^{14}\text{C}$ of SOC-derived respired CO₂. Consequently, our analysis and interpretation of $\Delta^{14}\text{C}$ of respired CO₂ are based on 80 of the 97 sites for which SIC contributions were limited (Extended Data Figs. 5 and 6).'

[Comment 3] Also, in the drylands, is there a depth component to the soils? How deep are the soils and could there be a significant contribution of older or fossil carbon in these ecosystems because there has not been the flux of water to significantly break down this organic flux from the bedrock. If the soils are quite thin and vegetation is not contributing to the OC pool, could fossil carbon contribute more significantly thus driving OC ages older, but without contributing to SOC storage from the terrestrial environment or inputs? There are new papers discussing this source and drylands might

be important when considering this flux of carbon into the ecosystems (Evans et al, 2025, Grant et al, 2023).

[Response] We thank the reviewer for raising this important point and for highlighting recent work on the role of rock-derived or petrogenic OC in soil carbon dynamics. Our study concentrates on topsoils (<10 cm depth). Deeper samples were not consistently available across regions (except for some Chinese sites), so assessing deep soil C dynamics and total profile depth is beyond the scope of this study. We are currently preparing a separate manuscript on depth profiles in Chinese dryland regions.

We acknowledge that petrogenic/fossil OC may contribute to old SOC in some drylands where fresh vegetation inputs are limited and parent material may contain ancient OC (Grant et al. 2023, Evans et al. 2025). However, for our topsoil samples, petrogenic/fossil OC is unlikely to be the major contributor to the bulk SOC, because bulk SOC $\Delta^{14}\text{C}$ values (-418.0‰ to -5.9‰) are far higher than ^{14}C -dead petrogenic OC (-1000‰). We estimated the potential fraction of fossil OC in bulk SOC using a two-end-member radiocarbon mass balance:

$$f_{\text{fossil}} * \Delta^{14}\text{C}_{\text{fossil}} + f_{\text{bio}} * \Delta^{14}\text{C}_{\text{bio}} = \Delta^{14}\text{C}_{\text{SOC}}$$

$$f_{\text{fossil}} + f_{\text{bio}} = 1$$

where $\Delta^{14}\text{C}_{\text{fossil}}$, $\Delta^{14}\text{C}_{\text{bio}}$, and $\Delta^{14}\text{C}_{\text{SOC}}$ represent the $\Delta^{14}\text{C}$ of fossil OC (-1000‰), $\Delta^{14}\text{C}$ of young OC from recent biospheric inputs (approximated as the $\Delta^{14}\text{C}$ of atmosphere CO_2 in the sampling year), and $\Delta^{14}\text{C}$ of bulk SOC, respectively. f_{fossil} and f_{bio} represent the fractions of fossil carbon and biospheric carbon in SOC. This yields a mean f_{fossil} of $16 \pm 1\%$ (mean \pm SE), indicating that fossil OC does not dominate SOC (Extended Data Table 4).

We admit that we do not know the original parent material of many of our soils, though several are loess that has been shown to contain C older than its depositional age (Wang et al. 2003). For those sites where soils contain SIC (including those developed on loess), the $\Delta^{14}\text{C}$ of SIC is always lower (older) than that of bulk SOC, showing that bulk SOC has a larger component of faster cycling C.

We have added an explanation of the potential sources of old C in lines 133–137: ‘While we cannot exclude the possibility of small amounts of ^{14}C -dead petrogenic OC (-1000‰), the higher $\Delta^{14}\text{C}$ values of bulk SOC (-418.0‰ to -5.9‰) prove that petrogenic OC can influence but does not dominate the old SOC pools, comprising at most 16% of the bulk SOC pool (Extended Data Table 4).’

[Comment 4] Additionally, one part of the discussion that is missing is where is the older portion of SOC coming from, you calculated that 23% of the CO_2 respired comes from older pools. Is there some thought on whether this is somehow from mineral associated OC that can be broken down or is it from POC that is not mineral associated?

In arid regions where could this older pool that is vulnerable to respiration and degradation likely stored until it is then destabilized?

[Response] We appreciate the reviewer’s constructive comment. To examine where ‘old’ respired C comes from, we considered two scenarios. In the first scenario, we assume that the $\Delta^{14}\text{C}$ of the old pool is represented by the $\Delta^{14}\text{C}$ of bulk SOC, yielding a high-end estimate in which the old pool contributes on average ~23% of the respired CO_2 (f_{old} , Extended Data Table 6). In the second scenario, we assume that the $\Delta^{14}\text{C}$ of the old pool is represented by ^{14}C -free petrogenic C (-1000‰), yielding a low-end estimate in which old carbon contributes ~4% of respired CO_2 ($f_{\text{old_petro}}$, Extended Data Table 6). While it is possible that there is indeed a limited amount of petrogenic C in our topsoils, we find it unlikely that this ‘inert’ C would contribute to microbial respiration in topsoils for four reasons:

(1) Petrogenic OC is generally treated as highly refractory and is expected to contribute mainly to old bulk SOC ages in deeper horizons, rather than to heterotrophic respiration (Copard et al. 2025).

(2) Studies in loess and other sediment-derived soils show substantial intermediate-aged, biospheric SOC components; on average about 40-50% of carbon in loess profiles was released at low temperatures and had chemical composition associated with plants dating from loess deposition (Wang et al. 2003). The residual fraction released at high temperatures was older but not radiocarbon-free (Wang et al. 2003).

(3) Ramped-combustion studies indicate that the majority of MAOC has intermediate ages (centuries to millennia), while much older fractions ($>10,000$ years), when present, account for only 1–4% of SOC even in settings where petrogenic inputs are expected (Stoner et al. 2023).

(4) A two-pool model with fast and slow pools but no inert pool can explain radiocarbon changes over time in many topsoils; only in deeper soils and in some cases (e.g., with fire inputs) is a third, ‘inert’ C pool required to explain radiocarbon time series (Trumbore 2009). As we have no independent measurements of petrogenic C across sites, we can only make arguments that the most likely combination is between these two extremes (4%-23%), depending on the ^{14}C content assumed for the ‘old’ component (4% if radiocarbon-free; 23% if equivalent to bulk SOC).

To further improve our estimates of the contribution of ‘old’ C to microbial respiration in dryland soils, we realized we could take advantage of available information on the POC and MAOC fractions of bulk SOC at the 41 BIODESERT sites we analyzed; these data are reported in a previous study (Díaz-Martínez et al. 2024). If we treat ‘POC’ and ‘MAOC’ as ‘fast’ and ‘slow’ cycling C pools, respectively, we obtain an intermediate estimate ($f_{\text{old_MAOC}} = 14\%$; Extended Data Table 6). In addition, $\Delta^{14}\text{C}$ of respired CO_2 is significantly related to both SOC fractions (Extended Data Fig. 3):

Extended Data Fig. 3 | Relationships between $\Delta^{14}\text{C}$ of respired CO_2 and SOC fractions. (a) $\Delta^{14}\text{C}$ of respired CO_2 vs. POC ($R^2 = 0.23$, $p < 0.01$, $n = 41$). (b) $\Delta^{14}\text{C}$ of respired CO_2 vs. MAOC ($R^2 = 0.26$, $p < 0.001$, $n = 41$). These patterns indicate that both POC and MAOC fractions can contribute to the respired CO_2 . Significance levels: ** $p < 0.01$, and *** $p < 0.001$.

Thus, a more reasonable explanation is that biospheric SOC spans a continuum of turnover times, reflecting physical protection (POC vs. MAOC) and variable mineral binding strengths (strong vs. weak). In contrast, we consider it unlikely that our results can be explained by only a small inert (^{14}C -dead) component with the remainder being ‘modern’. Unfortunately, all archived soils have now been used up, so we are unable to measure $\Delta^{14}\text{C}$ for POC and MAOC to assign the $\Delta^{14}\text{C}$ of respired CO_2 to one fraction.

Accordingly, we revised the ‘Estimating the contribution of old SOC to respired CO_2 ’ section in Methods (lines 376–413): ‘

Estimation of the contribution of old SOC to respired CO_2

The one-pool model indicates that the mean age of respired C (i.e., transit time) from bulk soils averages 520 ± 30 years (Fig. 1a), ranging from a few years up to 1,200 years. Because this is much younger than the mean age of bulk SOC ($2,100 \pm 140$ years, Fig. 1a), the respired CO_2 must originate from at least two C pools cycling on different timescales, with a faster pool contributing most of the respired CO_2 and a slower pool dominating the bulk SOC age. We assumed that the fast pool contains recently fixed C, and we considered two cases for the $\Delta^{14}\text{C}$ signature of the older C: one is the slow-cycling pool has the same $\Delta^{14}\text{C}$ of bulk SOC, and the other assuming the bulk SOC age represents a mixture of modern C and ^{14}C -free petrogenic C (-1000‰). The proportion of old C in soil respired CO_2 (f_{old}) was estimated by the following equations:

$$\Delta^{14}\text{C}_{\text{CO}_2} = \Delta^{14}\text{C}_{\text{old}} \times f_{\text{old}} + \Delta^{14}\text{C}_{\text{young}} \times f_{\text{young}} \quad (4)$$

$$f_{\text{old}} + f_{\text{young}} = 1 \quad (5)$$

where $\Delta^{14}\text{C}_{\text{CO}_2}$ represents the $\Delta^{14}\text{C}$ of total CO_2 released from the soil, and $\Delta^{14}\text{C}_{\text{young}}$ represents the $\Delta^{14}\text{C}$ of the atmosphere in the year of sample collection ($\Delta^{14}\text{C}_{\text{atmosphere}}$). If we assume $\Delta^{14}\text{C}_{\text{old}}$ equals the $\Delta^{14}\text{C}$ of bulk SOC, this provides the high-end estimate of the contribution of slow SOC to respiration. Under this assumption, the estimated mean f_{old} is $23 \pm 3\%$ (mean \pm SE), ranging from 6% to 69% across sites (Extended Data Table 6), even though this estimate involves uncertainties (e.g., the $\Delta^{14}\text{C}$ of the bomb-derived young C is most likely higher than that of atmospheric CO_2 in the sampling year). If we assume $\Delta^{14}\text{C}_{\text{old}}$ to represent ^{14}C -free petrogenic C, the estimated f_{old} (hereafter $f_{\text{old-petro}}$) is $4 \pm 0.4\%$, ranging from 1% to 12% (Extended Data Table 6). We consider the second assumption to be less likely than the first assumption (see Discussion). If bulk SOC contains mostly intermediate-aged C (e.g., average of centennial to millennial ages), a more reasonable assumption is that f_{old} is greater than 4%, but lower than 23%.

For the 41 sites from BIODESERT, we had data on ‘POC’ and ‘MAOC’ contents¹²; these fractions are often used to approximate ‘fast’ and ‘slow’ cycling C pools³¹ and can help us make a better approximation of f_{old} . We assumed that POC (mostly assumed to be fast-cycling) has modern $\Delta^{14}\text{C}$ values, and estimated $\Delta^{14}\text{C}_{\text{MAOC}}$ using a mass balance:

$$\text{bulk SOC } \Delta^{14}\text{C} = \Delta^{14}\text{C}_{\text{POC}} \times w_{\text{POC}} + \Delta^{14}\text{C}_{\text{MAOC}} \times w_{\text{MAOC}} \quad (6)$$

where $\Delta^{14}\text{C}_{\text{POC}}$ is assumed to reflect modern C ($\approx 0\%$), and w_{POC} and w_{MAOC} represent the fractions of POC and MAOC to SOC, respectively. Under these assumptions, $\Delta^{14}\text{C}_{\text{MAOC}}$ equals bulk $\Delta^{14}\text{C}/w_{\text{MAOC}}$. This estimated $\Delta^{14}\text{C}_{\text{MAOC}}$ was then used as the ‘old’ endmember ($\Delta^{14}\text{C}_{\text{old}}$) in the two-endmember mixing model (Equations 4–5) to quantify the contribution of MAOC to respiration. This yields a mean $f_{\text{old-MAOC}}$ of $14 \pm 2\%$, with values ranging from 3% to 53% (Extended Data Table 6).’

We then added the potential ‘old’ respired C source in discussion section (lines 145–152): ‘We believe that petrogenic OC is unlikely to be a dominant contributor to respired CO_2 , because it is highly refractory²⁸. Ramped combustion studies typically showed that mineral-associated organic C (MAOC) contains only a small ^{14}C -dead fraction²⁹. The relatively old respired CO_2 in drylands could more plausibly be explained by the continuum of ages spanned by biospheric SOC, reflecting physical protection (particulate organic C [POC] vs. MAOC) and variable mineral binding strengths (strong vs. weak)^{30,31} (Extended Data Fig. 3).’

Specific line-by line comments

[Comment 5] Line 75: Is there an accepted definition of drylands? A certain metric of aridity or precipitation? I think that would be very useful in the first paragraph for those who are not ecosystem scientists, it puts the actual metrics (rainfall or aridity in

perspective). I believe this is defined in the methods section, however, it is useful to mention it up front, so the reader can put it into context.

[Response] We thank the reviewer for this helpful suggestion. We have added the definition of drylands at the beginning of the introduction (lines 79–81): ‘Drylands, regions where the aridity index (the ratio of precipitation to potential evapotranspiration) is below 0.65¹, constitute the largest set of terrestrial biomes on Earth² and cover about 41% of the global land area³.’

[Comment 6] Line 110: This is a subtle difference, but it is 97 dryland sites correct? Or are there 97 unique dryland ecosystems?

[Response] We thank the reviewer for noticing this subtle difference. We have revised the sentence (line 115–117): ‘Here, we collected soils from 97 dryland sites spanning six continents and large environmental gradients (Extended Data Fig. 1).’

[Comment 7] Line 115: I find the wording of these hypotheses confusing. Could it be stated that reduced vegetation inputs drive older ages of SOC because there is less new OC incorporated into the soil? But wouldn't that be an effect of stronger mineral protection of the SOC that has already been stabilized? I would put the ‘with increasing aridity’ in the first part of the sentence hypothesis. The reduced aridity is driving the reduction in vegetation inputs, correct?

[Response] We agree that our original wording was confusing. We have revised the first hypothesis (lines 121–123): ‘We hypothesize that, with increasing aridity, reduced vegetation inputs and enhanced mineral protection lead to older ages of bulk SOC and respired CO₂.’

[Comment 8] Line 117: This next sentence of the hypothesis seems to say that with increased aridity you would expect the respired CO₂ to become even younger because of a preferential decomposition of recent C. It seems to me to contradict the next sentence, or maybe I am missing something, but could it be clearer if there was an added statement of the ¹⁴C of the evolved CO₂ from the respired CO₂ would in actuality decrease in amount and age as the fresh inputs are rapidly consumed, and the only SOC able for decomposition must come from other, older sources?

[Response] Thanks for the reviewer's insightful comments. Following the reviewer's suggestion, we have revised the second hypothesis to make this mechanism clearer (lines 123–128): ‘We further expect that, although the $\Delta^{14}\text{C}$ of both bulk SOC and respired CO₂ declines with increasing aridity, the $\Delta^{14}\text{C}$ difference between them will widen. This divergence reflects a growing decoupling between respiration, which is increasingly supported by a small pool of relatively fresh plant-derived inputs, and bulk

SOC, which is dominated by a larger, older C pool that contributes proportionally less to respiratory fluxes.’

[Comment 9] Line 139: Cite the aridity index calculation. Is it in the text or SI?

[Response] Thanks for this helpful comment. We now cite the standard international definition of the aridity index (precipitation/potential evapotranspiration) from the *World Atlas of Desertification* (Middleton and Thomas 1997) in line 585.

[Comment 10] Line 153: Could this be an artifact of the incubation? That the CO₂ evolved is somehow would not be that young in the field? Is there any discussion that in incubations there tends to be a preferential utilization of OC that was disturbed?

[Response] We thank the reviewer for raising this important concern. We acknowledge that laboratory incubations may not perfectly replicate field conditions. To minimize such artifacts, we did not grind the soils prior to incubation. Our results showed that soil texture (clay + silt) is not a dominant factor influencing the $\Delta^{14}\text{C}$ of either bulk SOC or respired CO₂. Previous work found that air-drying, rewetting, and storage had no substantial effects on the $\Delta^{14}\text{C}$ of respired CO₂ during incubation, indicating that these procedures did not mobilize C sources (Beem-Miller et al. 2021). We mentioned this in the Methods section (lines 340–342) as follows: ‘Incubation of archived soil samples has proven to be a promising tool for radiocarbon research, as air-drying, rewetting, and storage time have been shown to have relatively small effects on the $\Delta^{14}\text{C}$ of respired CO₂⁵⁶.’

[Comment 11] Line 156: What about DIC in arid soils? It seems to me that could contribute significant older carbon either from parent material carbonates or pedogenic carbonates. This hasn’t been mentioned yet in the text, and I would think in arid regions where there hasn’t been much weathering the soils would be full of base cations that could form carbonates and contribute to the CO₂ flux when re-wetted.

[Response] We thank the reviewer for raising this important point. We agree that DIC derived from parent material or pedogenic carbonates can influence the $\Delta^{14}\text{C}$ of respired CO₂ in dryland soils after rewetting (Barnard et al. 2020, Gallagher and Breecker 2020). We added a brief description of these mechanisms in the main text in response to your earlier comment 2 (lines 155–158).

We also added a new assumption in the Methods (lines 456–562): ‘We first assumed that carbonate was in equilibrium with CO₂ produced by SOC decomposition and treated all carbonate-derived CO₂ (solid SIC and its dissolution to DIC) as sharing a single carbonate end-member, and calculated the $\delta^{13}\text{C}$ of CO₂ in equilibrium with SIC (assuming a calcite–CO₂ fractionation of $\sim 9.6\text{‰}$ at 20 °C)⁶⁷, denoted as $\delta^{13}\text{C-SIC}_{\text{equilibrium}}$ in equation (7):

$$\delta^{13}\text{C-SIC}_{\text{equilibrium}} = \delta^{13}\text{C}_{\text{SIC}} - 9.6\text{‰} \quad (7)$$

where $\delta^{13}\text{C}_{\text{SIC}}$ represents the measured $\delta^{13}\text{C}$ of SIC in samples.'

[Comment 12] Line 157-161: This was the first mention of depth in this study. Were what depths the samples taken from or used in this study? Was it all topsoil? And what is the topsoil definition that was being used here? Can that be directly compared to the other topsoil values? And how would you expect this to change with depth?

[Response] We thank the reviewer for this helpful comment and agree that the definition of 'topsoil' and sampling depth should be stated more clearly. At the sites in Argentina, Australia, Iran, South Africa, Spain, and the United States, we followed the BIODESERT protocol and sampled the 0–7.5 cm soil layer. In the Chinese regions, surface soil was sampled from 0–10 cm to be consistent with existing national survey and long-term monitoring protocols. In our study, we use the term 'topsoil' to refer to these layers (0–7.5 or 0–10 cm), representing the biologically active surface SOC pool. We have clarified this definition and sampling depth in the Methods (lines 314–317).

The surface soil depth in Shi et al., 2020 is 0–30 cm; in this global synthesis, surface SOC ages were harmonized to this 0–30 cm layer by integrating studies that sampled various shallower depths (e.g., 0–10 cm, 0–20 cm) within the upper 30 cm. However, even when compared with the 0–30 cm surface layer used in global syntheses, the mean bulk SOC age in our dryland topsoils is still several times older than that reported for a wide range of other ecosystems, despite the limited availability of dryland data. We have added a brief clarification of this in the Results (line 173).

In addition, previous studies showed that $\Delta^{14}\text{C}$ values generally decreased with depth (Shi et al. 2020, Scheibe et al. 2023). We added this in lines 175–178: 'In addition, previous studies showed that bulk SOC ages generally increased with depth, due to reduced plant C inputs, lower microbial activity, and stronger mineral-associated stabilization in deeper horizons^{19,38}.'

[Comment 13] Line 167: This is not a modeled value correct? Or is it a 1-pool value? Can you clarify.

[Response] Thanks to the reviewer for pointing this out. The mean transit time is indeed derived from a one-pool model, in which case it equals the mean age derived from modeling the $\Delta^{14}\text{C}$ values of bulk SOC. We have clarified this on lines 181–183: 'Based on the $\Delta^{14}\text{C}$ values of respired CO_2 and a one-pool model, the mean transit time was estimated to be 520 ± 30 years (Fig. 1a; Extended Data Fig. 6; Extended Data Table 7).'

[Comment 14] Line 175: Do you mean the estimates for only drylands broken out from the ESM models or drylands compared to entire biomes from the ESM model?

[Response] We thank the reviewer for this helpful comment. In this sentence, we compare our dryland radiocarbon-based transit time estimates with the turnover times reported for dryland biomes in ESM-based and incubation-based studies, rather than with global averages across entire biomes. To make this clearer, we have revised lines 188–190 as follows: ‘Our estimates are also orders of magnitude higher than the year-to-decade turnover times of the fast pool and even the bulk SOC pool in dryland biomes as estimated by ESMs and incubation experiments^{25,39}.’

[Comment 15] Line 202: As mentioned before, can you briefly explain/define the aridity index?

[Response] Thanks for the helpful comment. We now add the calculation and definition of aridity and aridity index on lines 199–200.

[Comment 16] Line 258: I would argue other regions as well. What about the combination of comparing other fractions in these regions to ¹⁴C incubations or different compound classes in dryland ecosystems. Would or should there be an effort to attempt measure different compounds that could accurately assess the fast-cycling pool of soil carbon, without some of the draw backs of incubations?

[Response] Thanks for this insightful suggestion. We agree that combining $\Delta^{14}\text{C}$ measurements of bulk SOC and respired CO_2 provides valuable constraints on SOC turnover not only in dryland regions but also in other ecosystems globally. We now revised lines 260–262 as follows: ‘Together, these results underscore the importance of combining $\Delta^{14}\text{C}$ analysis of bulk SOC and respired CO_2 to derive more accurate estimates for constraining model simulations’

We also highlight that measuring $\Delta^{14}\text{C}$ in different soil fractions could further improve characterization of fast- and slow-cycling SOC pools. We added the following sentence in lines 257–260: ‘Future $\Delta^{14}\text{C}$ measurements of chemically or physically defined fractions (e.g., POC and MAOC), as well as compound-specific analysis (e.g., lignin phenols, amino sugars, and black carbon), could further improve characterization of fast- and slow-cycling SOC pools.’

[Comment 17] Line 383: This wording is confusing. It says there is a one-pool steady state model, but then the next sentence explains this assumes there are two-pool represented by what you measured? Can you clarify this sentence?

[Response] We thank the reviewer for pointing out this confusing wording. We conceptualized SOC as two effective pools, but each pool is represented by its own one-pool, steady-state radiocarbon model: one fitted to bulk SOC $\Delta^{14}\text{C}$ (a ‘slow’ pool) and one fitted to $\Delta^{14}\text{C}$ of respired CO_2 (a ‘fast’ pool). To avoid confusion, we have revised the text in lines 415–421: ‘The mean age of bulk SOC and the mean transit time of soil

respired C were estimated from $\Delta^{14}\text{C}_{\text{sample}}$ using two separate one-pool steady-state models: one fitted to the $\Delta^{14}\text{C}$ of bulk SOC and one fitted to the $\Delta^{14}\text{C}$ of respired CO_2 . In this framework, bulk SOC is treated as a ‘slow’ pool representing the majority of SOC mass, whereas respired CO_2 is derived from a ‘fast’ pool representing a small fraction of total SOC that is most readily decomposed¹⁶. Each pool is represented by its own one-pool model and is assumed to be homogeneous and at steady state (i.e., not accumulating or losing C).’

[Comment 18] Line 462: I do think some of this discussion of soil inorganic carbon needs to be in the main text.

[Response] Thanks for the reviewer’s helpful comment. We already added the discussion of SIC in the main text (see response to Comment 2).

References

- Barnard, R. L., S. J. Blazewicz, and M. K. Firestone. 2020. Rewetting of soil: Revisiting the origin of soil CO₂ emissions. *Soil Biology & Biochemistry* **147**.
- Beem-Miller, J., M. Schrupf, A. M. Hoyt, G. Guggenberger, and S. Trumbore. 2021. Impacts of drying and rewetting on the radiocarbon signature of respired CO₂ and implications for incubating archived soils. *Journal of Geophysical Research-Biogeosciences* **126**:e2020JG006119.
- Berdugo, M., M. Delgado-Baquerizo, S. Soliveres, R. Hernández-Clemente, Y. C. Zhao, J. J. Gaitan, N. Gross, H. Saiz, V. Maire, A. Lehman, M. C. Rillig, R. V. Solé, and F. T. Maestre. 2020. Global ecosystem thresholds driven by aridity. *Science* **367**:787-790.
- Copard, Y., C. Hatté, L. Cécillon, Y. Colin, P. Barré, C. Chenu, and S. Cornu. 2025. Soil carbon dynamics reshaped by ancient carbon quantification. *Global change biology* **31**:e70482.
- Díaz-Martínez, P., F. T. Maestre, E. Moreno-Jiménez, M. Delgado-Baquerizo, D. J. Eldridge, H. Saiz, N. Gross, Y. Le Bagousse-Pinguet, B. Gozalo, and V. Ochoa. 2024. Vulnerability of mineral-associated soil organic carbon to climate across global drylands. *Nature Climate Change* **14**:976-982.
- Eldridge, D. J., J. Y. Ding, J. Dorrough, M. Delgado-Baquerizo, O. Sala, N. Gross, Y. Le Bagousse-Pinguet, M. Mallen-Cooper, H. Saiz, S. Asensio, V. Ochoa, B. Gozalo, E. Guirado, M. García-Gómez, E. Valencia, J. Martínez-Valderrama, C. Plaza, M. Abedi, N. Ahmadian, R. J. Ahumada, J. M. Alcántara, F. Amghar, L. Azevedo, F. Ben Salem, M. Berdugo, N. Blaum, B. Boldgiv, M. Bowker, D. Bran, C. Bu, R. Canessa, A. P. Castillo-Monroy, I. Castro, P. Castro-Quezada, S. Cesarz, R. Chibani, A. A. Conceicao, A. Darrouzet-Nardi, Y. C. Davila, B. Deak, P. Diaz-Martínez, D. A. Donoso, A. D. Dougill, J. Duran, N. Eisenhauer, H. Ejtehadi, C. I. Espinosa, A. Fajardo, M. Farzam, A. Foronda, J. Franzese, L. H. Fraser, J. Gaitán, K. Geissler, S. L. Gonzalez, E. Gusman-Montalvan, R. M. Hernandez, N. Hölzel, F. M. Hughes, O. Jadan, A. Jentsch, M. C. Ju, K. F. Kaseke, M. Köbel, A. Lehmann, P. Liancourt, A. Linstadter, M. A. Louw, Q. H. Ma, M. Mabaso, G. Maggs-Kolling, T. P. Makhalanyane, O. M. Issa, E. Marais, M. McClaran, B. Mendoza, V. Mokoka, J. P. Mora, G. Moreno, S. Munson, A. Nunes, G. Oliva, G. R. Onatibia, B. Osborne, G. Peter, M. Pierre, Y. Pueyo, R. E. Quiroga, S. Reed, A. Rey, P. Rey, V. M. R. Gómez, V. Rolo, M. C. Rillig, P. C. Le Roux, J. C. Ruppert, A. Salah, P. J. Sebei, A. Sharkhuu, I. Stavi, C. Stephens, A. L. Teixido, A. D. Thomas, K. Tielbörger, S. T. Robles, S. Travers, O. Valko, L. van den Brink, F. Velbert, A. Von Hessberg, W. Wamiti, D. L. Wang, L. X. Wang, G. M. Wardle, L. Yahdjian, E. Zaady, Y. M. Zhang, X. B. Zhou, and F. T. Maestre. 2024. Hotspots of biogeochemical activity linked to aridity and plant traits across global drylands. *Nature Plants* **10**.
- Evans, D. L., S. Doetterl, N. Gallarotti, E. Georgiadis, S. Nabhan, S. H. Wartenweiler, T. M. Y. Rhyner, B. V. A. Mittelbach, T. I. Eglinton, J. Hemingway, and T. M. Blattmann. 2025. The Known Unknowns of Petrogenic Organic Carbon in Soils. *Agu Advances* **6**.

- Gallagher, T. M., and D. O. Breecker. 2020. The obscuring effects of calcite dissolution and formation on quantifying soil respiration. *Global Biogeochemical Cycles* **34**.
- Grant, K. E., R. G. Hilton, and V. V. Galy. 2023. Global patterns of radiocarbon depletion in subsoil linked to rock-derived organic carbon. *Geochemical Perspectives Letters* **25**:36-40.
- Maestre, F. T., B. M. Benito, M. Berdugo, L. Concostrina-Zubiri, M. Delgado-Baquerizo, D. J. Eldridge, E. Guirado, N. Gross, S. Kéfi, Y. Le Bagousse-Pinguet, R. Ochoa-Hueso, and S. Soliveres. 2021. Biogeography of global drylands. *New Phytologist* **231**:540-558.
- Maestre, F. T., D. J. Eldridge, N. Gross, Y. Le Bagousse-Pinguet, H. Saiz, B. Gozalo, V. Ochoa, and J. J. Gaitán. 2022a. The BIODESERT survey: assessing the impacts of grazing on the structure and functioning of global drylands. *Web Ecology* **22**:75-96.
- Maestre, F. T., D. J. Eldridge, S. Soliveres, S. Kéfi, M. Delgado-Baquerizo, M. A. Bowker, P. García-Palacios, J. Gaitán, A. Gallardo, and R. Lázaro. 2016. Structure and functioning of dryland ecosystems in a changing world. *Annual review of ecology, evolution, and systematics* **47**:215-237.
- Maestre, F. T., Y. Le Bagousse-Pinguet, M. Delgado-Baquerizo, D. J. Eldridge, H. Saiz, M. Berdugo, B. Gozalo, V. Ochoa, E. Guirado, and M. García-Gómez. 2022b. Grazing and ecosystem service delivery in global drylands. *Science* **378**:915-920.
- Maestre, F. T., R. Salguero-Gómez, and J. L. Quero. 2012. It is getting hotter in here: determining and projecting the impacts of global environmental change on drylands Introduction. *Philosophical Transactions of the Royal Society B-Biological Sciences* **367**:3062-3075.
- Middleton, N., and D. Thomas. 1997. *World atlas of desertification*. Arnold for UNEP, London.
- Scheibe, A., C. A. Sierra, and M. Spohn. 2023. Recently fixed carbon fuels microbial activity several meters below the soil surface. *Biogeosciences* **20**:827-838.
- Schlesinger, W. H., and A. M. Pilmanis. 1998. Plant-soil Interactions in Deserts. *Biogeochemistry* **42**:169-187.
- Shi, Z., S. D. Allison, Y. J. He, P. A. Levine, A. M. Hoyt, J. Beem-Miller, Q. Zhu, W. R. Wieder, S. Trumbore, and J. T. Randerson. 2020. The age distribution of global soil carbon inferred from radiocarbon measurements. *Nature Geoscience* **13**:555-559.
- Stoner, S., S. E. Trumbore, J. A. González-Pérez, M. Schrumpf, C. A. Sierra, A. M. Hoyt, O. Chadwick, and S. Doetterl. 2023. Relating mineral-organic matter stabilization mechanisms to carbon quality and age distributions using ramped thermal analysis. *Philosophical Transactions of the Royal Society a-Mathematical Physical and Engineering Sciences* **381**.
- Sun, J., G. W. Cheng, and W. P. Li. 2013. Meta-analysis of relationships between environmental factors and aboveground biomass in the alpine grassland on the Tibetan Plateau. *Biogeosciences* **10**:1707-1715.

- Trumbore, S. 2009. Radiocarbon and soil carbon dynamics. *Annual Review of Earth and Planetary Sciences* **37**:47-66.
- Wang, H., K. C. Hackley, S. V. Panno, D. D. Coleman, J. C. L. Liu, and J. Brown. 2003. Pyrolysis-combustion ¹⁴C dating of soil organic matter. *Quaternary Research* **60**:348-355.